# Knowledge-Level Consistency Reinforcement Learning: Dual-Fact Alignment for Long-Form Factuality

## Abstract

Hallucination and factuality deficits remain key obstacles to the reliability of large language models (LLMs) in long-form generation. Existing reinforcement learning from human feedback (RLHF) frameworks primarily rely on preference rewards, yet they often overlook the model's internal knowledge boundaries, exacerbating the so-called "hallucination tax". To address this challenge, we propose **K**nowledge-**L**evel **C**onsistency Reinforcement Learning **F**ramework (**KLCF**), a novel framework that focuses on the knowledge consistency between the policy model's expressed knowledge and the base model's parametric knowledge, and introduces a Dual-Fact Alignment mechanism to jointly optimize factual recall and precision. Specifically, KLCF leverages pretrained knowledge boundaries to construct fact checklist, guiding online reinforcement learning to improve factual coverage and recall; simultaneously, it trains a self-assessment module based on the base model's internal knowledge to enhance factual precision during generation. Unlike prior methods that rely on external retrieval or heavy verification, our reward design is fully online external-knowledge-free and lightweight, making KLCF efficient and easily scalable to large-scale training. Experimental results demonstrate that KLCF substantially improves factuality metrics across multiple long-form benchmarks and effectively alleviates model hallucinations.

## 1 Introduction

Large language models (LLMs) have shown strong performance across a wide range of NLP tasks, such as question answering (Lewis et al., 2020; Li et al., 2024; Xu et al., 2024), content summarization (Zhang et al., 2024c; Gupta et al., 2025), and complex reasoning (Wei et al., 2022; Shao et al., 2024; Zhang et al., 2025b). However, hallucinations, in which the model generates outputs that conflict with established facts, remain a central barrier to reliable deployment (Huang et al., 2025a; Wang et al., 2024). The problem is especially acute in long-form generation, where spurious statements introduced early can cascade via a "snowball effect" (Zhang et al., 2023; Ji et al., 2023; Huang et al., 2025a), progressively amplifying errors and undermining the credibility of the final response. Furthermore, existing reinforcement learning from human feedback (RLHF) frameworks (Bai et al., 2022) rely on preference-based rewards yet often overlook the model's parametric-knowledge boundary. This may cause misalignment and encourage models to fabricate facts beyond their knowledge boundaries, exacerbating the so-called "hallucination tax" (Cotra, 2021; Sharma et al., 2023).

To evaluate and address the factuality and hallucination issues in long-form generation, a commonly employed approach is to decompose the model's output into atomic facts and verifies them through external retrieval. The early work FActScore (Min et al., 2023) and FacTool (Chern et al., 2023) established this paradigm. Subsequently, methods like SAFE (Wei et al., 2024) and VeriScore (Song et al., 2024) further refined the fact extraction and verification pipeline. Methods like FactTune-FS (Tian et al., 2023), FLAME (Lin et al., 2024a) and FactAlign (Huang & Chen, 2024) use evaluators like FActScore to provide factuality alignment signals through external retrieval, successfully reducing hallucinations to some extent. However, most of these approaches are still confined to offline Reinforcement Learning (RL) scenarios. The low efficiency of external retrieval-based verification making these methods difficult to scale for large-scale online RL applications. In terms of

Figure 1: Conceptual illustration of the KLCF alignment motivation. Conventional long-form factuality alignment methods reduce hallucinations by shrinking the Expressed Knowledge at the cost of coverage, while KLCF aims to expand the intersection of the Expressed Knowledge and the Parametric knowledge acquired from pre-training.

effectiveness, the main limitations of these approaches include a narrow focus on factual precision without considering factual recall, often leading models to favor conservative responses.

As illustrated in Fig. 1, based on the principle that LLMs' parametric knowledge originates from pre-training (Petroni et al., 2019; Haeun Yu, 2024) and the aligned model learns how to express it, we propose that the core objective of long-form factuality alignment is to increase the knowledge-level consistency between the aligned model's "Expressed Knowledge" (i.e., the knowledge contained within the policy model's generated content) and the base model's "Parametric Knowledge" (i.e., the internal knowledge encoded during pre-training). This alignment objective is twofold: maximizing the factual recall from the parametric knowledge while minimizing the generation of content beyond its knowledge boundary. Previous work (Kadavath et al., 2022; Zhang et al., 2024a) has shown that this alignment objective can potentially be achieved solely by relying on internal knowledge to provide the reward signal.

Building on this motivation, we introduce the **K**nowledge-**L**evel **C**onsistency Reinforcement Learning **F**ramework (**KLCF**), designed to align models through knowledge-level consistency in an online reinforcement learning setting. KLCF incorporates a Dual-Fact Alignment mechanism, operationalizing this alignment with two complementary knowledge-level consistency rewards (KLC rewards): a Checklist-Based Consistency Reward (Checklist Reward), which supervises the generation against a pre-constructed checklist derived from the pre-trained model's parametric knowledge to enhance factual coverage and recall; and a Confidence-Based Truthfulness Reward (Truthfulness Reward), which employs a self-assessment module to perform fine-grained evaluation of truthfulness, preventing the aligned model's responses from generating untrue statements that exceed its knowledge boundary. This synergy encourages the model to be both expressive and factual reliable, alleviating the hallucination-conservatism trade-off. Our main contributions are as follows:

- We propose KLCF, a novel RL framework that introduces knowledge-level consistency as a core objective, aiming to reduce hallucinations by aligning the model's expressed knowledge with its parametric knowledge.

- We design a Dual-Fact Alignment mechanism to achieve this consistency, which jointly optimizes for factual recall and precision by providing dual rewards for RL.

- We develop a lightweight, online external-knowledge-free reward system that eliminates external knowledge retrieval and enables scalable online RL training.

- Extensive experiments show KLCF achieves consistent gains across diverse benchmarks, model scales, and reasoning modes.

## 2 KNOWLEDGE-LEVEL CONSISTENCY REINFORCEMENT LEARNING

### 2.1 OVERVIEW

As illustrated in the left part of Fig 2, the Knowledge-Level Consistency Reinforcement Learning Framework (KLCF) is designed to align the model's expressed knowledge with its parametric knowledge during online reinforcement learning. This alignment is driven by our knowledge-level consistency rewards (Checklist Reward and Truthfulness Reward). In contrast to prior methods (Fig 2, right) that rely on costly external knowledge retrieval for real-time fact verification during RL, our approach places greater emphasis on offline data preparation. This phase constructs a verified factual checklist and training data for a truthfulness reward model from the base model's knowl-

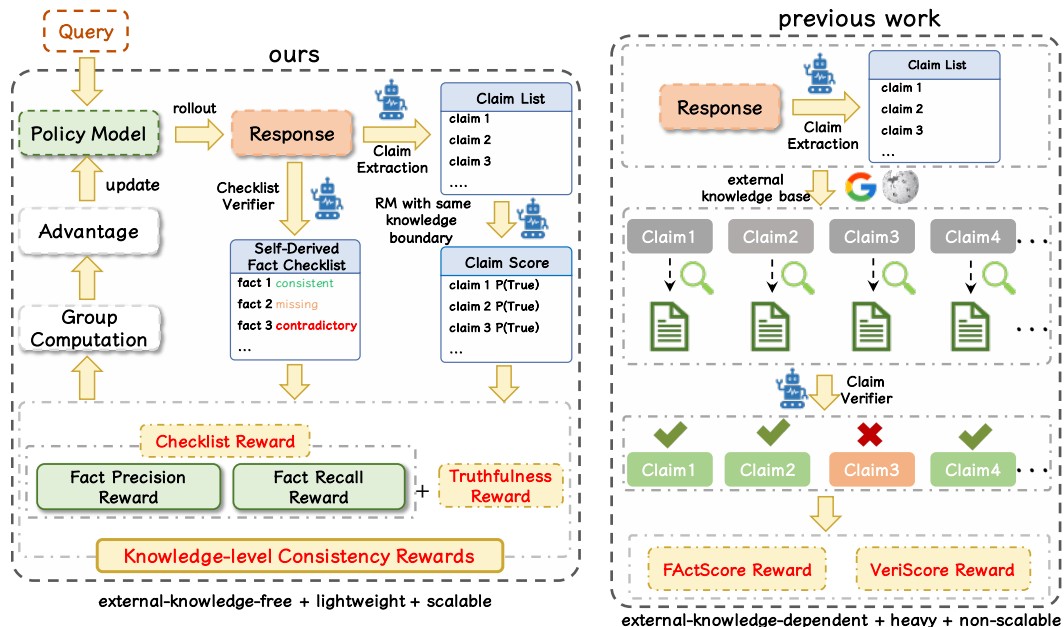

Figure 2: KLCF framework (Left) vs. Previous work (Right). Unlike previous methods that rely on costly online external knowledge retrieval for real-time verification, our framework achieves dual-fact alignment through knowledge-level consistency rewards, which are computed efficiently using offline-prepared resources—a factual checklist and a truthfulness reward model—enabling scalable RL training without external dependencies.

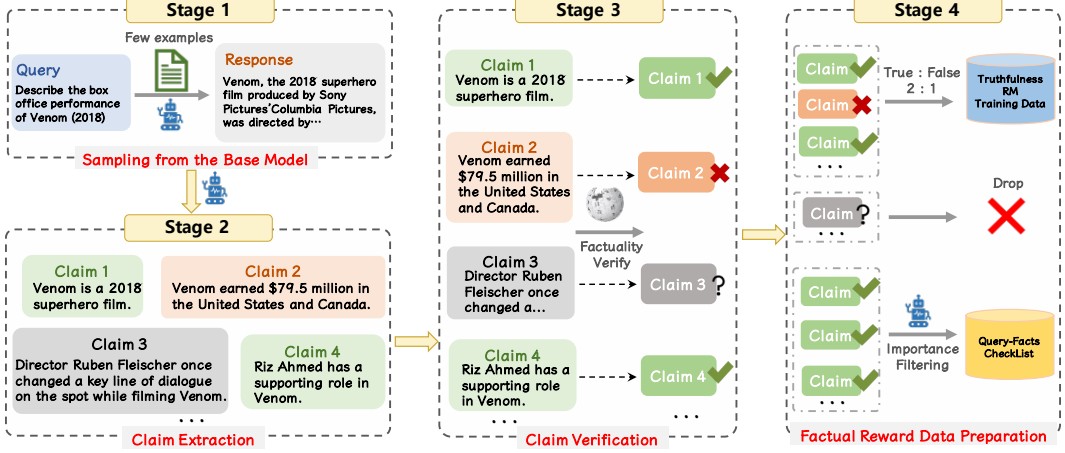

Figure 3: Offline data preparation pipeline. The process constructs the essential resources for knowledge-level consistency rewards—a factual checklist and truthfulness reward model training data—by extracting and verifying claims from the base model's responses.

edge boundary, enabling the efficient computation of these knowledge-level consistency rewards. In the following, we first detail the offline data preparation process (Section 2.2). We then describe the core knowledge-level consistency rewards (Section 2.3.1) and auxiliary rewards (Section 2.3.2), followed by the policy optimization procedure (Section 2.3.4).

## 2.2 DATA PREPARATION FOR FACTUAL ALIGNMENT

We construct our training data from three sources, namely ELI5 (Fan et al., 2019), LongFact-Gen, and LongWiki-Gen, to ensure diversity and factual grounding. LongFact-Gen is regenerated using

GPT-4.1 based on the original LongFact (Wei et al., 2024) prompts, and LongWiki-Gen is built by applying the methodology from (Bang et al., 2025) to the GoodWiki corpus (Choi, 2023), ensuring no overlap with their respective test sets. The overall pipeline for processing these datasets, illustrated in Fig. 3, consists of four sequential stages detailed below. We provide the exact configuration and prompts for all components in Appendix C.1 and Appendix E.

**Stage 1: Sampling from the Base Model.** To probe the knowledge boundary of the base model $\pi_{\text{base}}$, we prompt it to generate multiple responses for each query $q_i$ in a few-shot setting. Specifically, for each query, we perform $\nu$ independent samplings (where $\nu = 8$) with a high temperature setting to elicit a more diverse and comprehensive set of factual statements that $\pi_{\text{base}}$ is capable of generating. Formally, for a query set $\mathcal{Q} = \{q_1, q_2, ..., q_N\}$, we obtain a set of responses $\mathcal{O} = \{o_{i,1}, o_{i,2}, ..., o_{i,\nu}\}_{i=1}^{N}$, where $o_{i,j} \sim \pi_{\text{base}}(\cdot | q_i, \mathcal{C})$ and $\mathcal{C}$ represents the few-shot context. These responses serve as the raw material for subsequent factual mining.

**Stage 2: Claim Extraction.** After obtaining the base model's responses for each query in the dataset, we extract verifiable factual information from them. We train a lightweight claim extraction model $f_{\text{extract}}$ to parse each response $o_{i,j}$ and output a set of claims. For each query $q_i$, we define $C(o_i)$ as the union of claims extracted from all $\nu$ responses:

$$C(o_i) = \bigcup_{j=1}^{\nu} f_{\text{extract}}(o_{i,j}) = \{c_{i,1}, c_{i,2}, ..., c_{i,M_i}\} \tag{1}$$

This step converts unstructured text into a structured set of claims $\mathcal{C}_{\text{all}} = \bigcup_{i=1}^{N} C(o_i)$, which is essential for precise verification.

**Stage 3: Claim Verification.** Each claim $c \in \mathcal{C}_{\text{all}}$ undergoes a verification process against a local Wiki20250716 knowledge index, built from a processed Wikipedia dump and indexed via (Chen et al., 2017; Wang et al., 2022). For each claim, the top-$k$ (where $k = 10$) most relevant documents $D(c)$ are retrieved. The claim and retrieved context are then evaluated by the Qwen2.5-72B-Instruct verifier model $f_{\text{verify}}$, which assigns a veracity label:

$$l = f_{\text{verify}}(c, D(c)), \quad l \in \{\text{SUPPORT}, \text{REFUTE}, \text{NOT ENOUGH INFO}\} \tag{2}$$

This ensures all claims are assessed against reliable external knowledge, producing verified labels for reward modeling.

**Stage 4: Factual Reward Data Preparation.** The verified claims are curated to construct the two core components for our dual-fact alignment mechanism, which aims to achieve knowledge-level consistency by jointly optimizing factual recall and precision. First, the factual checklist $\Lambda(q_i)$ for each query $q_i$ is formally constructed through the following aggregation and refinement process:

$$\Lambda(q_i) = f_{\text{filter}}\left(\{c \in C(o_i) \mid f_{\text{verify}}(c, D(c)) = \text{SUPPORT}\}\right) \tag{3}$$

where $f_{\text{filter}}$ operation applies a prompt-based refinement to remove duplicates and low-importance facts, ultimately producing a concise knowledge coverage blueprint. Second, the truthfulness reward model training dataset is formally defined as:

$$\mathcal{D}_{\text{truth}} = \{(c, \text{True}) \mid c \in \mathcal{C}'_{\text{SUPPORT}}\} \cup \{(c, \text{False}) \mid c \in \mathcal{C}'_{\text{REFUTE}}\} \tag{4}$$

where $\mathcal{C}'_{\text{SUPPORT}}$ and $\mathcal{C}'_{\text{REFUTE}}$ correspond to the carefully sampled subsets of SUPPORT and REFUTE-labeled claims from Stage 3, maintaining a 2:1 positive-to-negative ratio where each pair $(c, \text{True})$ or $(c, \text{False})$ serves as a training instance for the reward model to estimate $p(\text{True} \mid c)$. These two resources directly enable the computation of our knowledge-level consistency rewards: the checklist facilitates the Checklist Reward to enhance factual coverage, while the balanced claim set trains a model to provide the Truthfulness Reward for improved factual precision during online generation.

## 2.3 REINFORCEMENT LEARNING FRAMEWORK

This section details the core mechanism of KLCF: the reward design and policy optimization that drive knowledge-level consistency. We first formalize the key components involved in the online reinforcement learning process. For a given query $q_i$, the policy model generates a structured response $o_i = <\text{think}>\mathcal{T}_i</\text{think}> <\text{answer}>\mathcal{A}_i</\text{answer}>$. Additionally, a fact checklist $\Lambda(q_i)$ is predefined for each query $q_i$. The final reward is denoted as $R(o_i)$.

### 2.3.1 KNOWLEDGE-LEVEL CONSISTENCY REWARDS

We now introduce the core of our framework, the knowledge-level consistency rewards, which consist of the Checklist-Based Consistency Reward (Checklist Reward) and the Confidence-Based Truthfulness Reward (Truthfulness Reward).

**Checklist Reward.** The Checklist Reward evaluates the factual coverage of model responses. Using a factual verification prompt (Prompt 7), each item in the checklist $\Lambda(q_i)$ is compared to the response $\mathcal{A}_i$ and classified as **Consistent**, **Contradictory**, or **Missing**. This fine-grained classification enables the definition of two key rewards: fact recall reward and fact precision reward.

- **Fact recall reward** measures the proportion of relevant facts that are correctly included in the response $\mathcal{A}_i$, computed as the number of consistent facts divided by the total number of facts in the checklist:

$$R_{\text{recall}}(\mathcal{A}_i) = \frac{N_{\text{consistent}}(i)}{N_{\text{consistent}}(i) + N_{\text{contradictory}}(i) + N_{\text{missing}}(i)} \quad (5)$$

  where $N_{\text{consistent}}(i)$, $N_{\text{contradictory}}(i)$, and $N_{\text{missing}}(i)$ denote counts of checklist facts judged as Consistent, Contradictory, or Missing, respectively. Fact recall measures the percentage of checklist items correctly covered by the model's response, reflecting its completeness.

- Next, we also define **fact precision reward**, which represents the proportion of all claims mentioned in the model's responses that are correct:

$$R_{\text{precision}}(\mathcal{A}_i) = \frac{N_{\text{consistent}}(i)}{N_{\text{consistent}}(i) + N_{\text{contradictory}}(i)} \quad (6)$$

To unify the fact recall reward and fact precision reward into a comprehensive reward signal, we introduce the checklist reward as a weighted harmonic balance defined by:

$$R_{\text{checklist}}(\mathcal{A}_i) = \frac{1}{3} \times R_{\text{recall}}(\mathcal{A}_i) + \frac{2}{3} \times R_{\text{precision}}(\mathcal{A}_i) \quad (7)$$

**Truthfulness Reward.** While the fact precision reward measures precision specifically against the pre-defined checklist, we further introduce a Truthfulness Reward to enhance the overall factual truthfulness of the generated response. This reward is implemented via a specialized reward model that evaluates the truthfulness of claims in the generated response. Specifically, a lightweight claim extraction model first extracts a set of verifiable atomic claims from $\mathcal{A}_i$, denoted as $C(\mathcal{A}_i) = \{c_{i,1}, c_{i,2}, ...\}$. Each claim $c_{i,j} \in C(\mathcal{A}_i)$ is then assessed by the truthfulness reward model to estimate its probability of being true, $p(\text{True} \mid c_{i,j})$. The overall truthfulness reward is computed as the average of these probabilities:

$$R_{\text{truth}}(\mathcal{A}_i) = \frac{1}{C(\mathcal{A}_i)} \sum_{j=1}^{|C(\mathcal{A}_i)|} p(\text{True} \mid c_{i,j}) \quad (8)$$

We also propose a variant of the truthfulness reward that reduces noise by leveraging the pre-verified factual checklist $\Lambda(q_i)$ as high-confidence prior knowledge. Instead of evaluating all claims in the response with the reward model—which may introduce noise—we skip claims already covered by $\Lambda(q_i)$, since their factuality is assured. Specifically, all items in $\Lambda(q_i)$ are concatenated into a pseudo-response. The claims $C(\mathcal{A}_i)$ from the model response are then verified against this pseudo-response using the same protocol as the checklist reward. The variant reward $R_{\text{truth}}^{\text{variant}}$ is computed only over the subset of claims $C_M(\mathcal{A}_i)$ marked as **Missing**:

$$R_{\text{truth}}^{\text{variant}}(\mathcal{A}_i) = \mathbb{I}_{|C_M(\mathcal{A}_i)|>0} \cdot \frac{1}{|C_M(\mathcal{A}_i)|} \sum_{j=1}^{|C_M(\mathcal{A}_i)|} p(\text{True} \mid c_{i,j}) \quad (9)$$

where $\mathbb{I}$ is indicator function.

**Discussion.** Our dual reward mechanism effectively narrows the gap between the policy model's expressed knowledge and its internal parametric knowledge from two complementary dimensions,

increasing their intersection. Specifically, the **checklist reward** externally constrains the model by leveraging a verifiable fact set from the base model's knowledge boundary. This aims to improve fact recall and fact precision on a specific set, encouraging the model to express its internal knowledge more fully. Concurrently, the **truthfulness reward** works internally, using a self-assessment module trained on the same parametric knowledge to judge the confidence of the generated content. This aims to suppress the generation of uncertain or fabricated information that lies beyond the model's knowledge boundary, ensuring factual precision. The synergy of these two rewards prompts the model to confidently express what it "knows" while cautiously avoiding what it "does not know", thereby systematically achieving knowledge-level consistency reinforcement.

### 2.3.2 Auxiliary Rewards

To ensure the overall quality of generated text beyond factuality and prevent potential degradation in readability or adherence to instructions, we introduce three auxiliary rewards.

**General Reward.** As we perform RL directly from the base model—bypassing SFT—a standard KL penalty is inapplicable. To prevent the policy from deviating into low-quality outputs, we employ a general reward $R_g(\mathcal{A}_i)$ from Skywork-Reward-V2-Llama-3.2-1B (Liu et al., 2025) to incentivize responses that align with human preference.

**Format Reward.** We introduce a Format Reward $R_f$ to enforce structured output in our thinking model. The output must encapsulate reasoning within <think></think> tags and the final answer within <answer></answer> tags:

$$R_f(o_i) = \begin{cases} 0, & \text{if } o_i \text{ has valid format} \\ -1, & \text{otherwise} \end{cases} \tag{10}$$

**Length Penalty.** To ensure the conciseness and information density of long-form factual responses, and to prevent the model from generating redundant content, we introduce a Length Penalty (Yu et al., 2025). Let $L_m$ denote the maximum allowed length threshold and $L_c$ ($L_c < L_m$) be a predefined critical length value. The Length Penalty $R_l$ is defined piecewise as follows:

$$R_l(\mathcal{A}_i) = \begin{cases} 0, & |\mathcal{A}_i| \leq L_m - L_c \\ \dfrac{(L_m - L_c) - |\mathcal{A}_i|}{L_c}, & L_m - L_c < |\mathcal{A}_i| \leq L_m \\ -1, & |\mathcal{A}_i| > L_m \end{cases} \tag{11}$$

### 2.3.3 Reward Combination

The final reward function $R(o_i)$ for response $o_i$ can be uniformly expressed in the following piecewise form:

$$R(o_i) = \begin{cases} R_{\text{checklist}}(\mathcal{A}_i) + 0.1 \cdot R_g(\mathcal{A}_i) + R_l(\mathcal{A}_i) + R_f(o_i), & \text{not using } R_{\text{truth}} \\ R_{\text{truth}}(\mathcal{A}_i) + 0.1 \cdot R_g(\mathcal{A}_i) + R_l(\mathcal{A}_i) + R_f(o_i), & \text{not using } R_{\text{checklist}} \\ R_{\text{fact}}(\mathcal{A}_i) + 0.1 \cdot R_g(\mathcal{A}_i) + R_l(\mathcal{A}_i) + R_f(o_i), & \text{both} \end{cases} \tag{12}$$

where $R_{\text{fact}}(\mathcal{A}_i)$ is defined as:

$$R_{\text{fact}}(\mathcal{A}_i) = \kappa \cdot R_{\text{recall}}(\mathcal{A}_i) + \lambda \cdot R_{\text{precision}}(\mathcal{A}_i) + \mu \cdot R_{\text{truth}}(\mathcal{A}_i) \tag{13}$$

where $\kappa$, $\lambda$, and $\mu$ are hyperparameters between 0 and 1, satisfying $\kappa + \lambda + \mu = 1$. We provide a detailed discussion of their selection in the Section 3.3.

### 2.3.4 Policy Optimization

To optimize the model towards the reward objectives defined in the previous section, we employ Group Relative Policy Optimization (GRPO) (Shao et al., 2024). For a given question-answer pair $(q, a)$, the policy $\pi_{\theta_{\text{old}}}$ generates a group of $G$ responses $G\{o_i\}_{i=1}^G$. The advantage of the $i$-th response is then obtained by normalizing the group-level rewards $\{R(o_i)\}_{i=1}^G$, and the importance sampling ratio between the updated and behavior policy is defined as:

$$\hat{A}_{i,t} = \frac{R(o_i) - \text{mean}(\{R(o_i)\}_{i=1}^G)}{\text{std}(\{R(o_i)\}_{i=1}^G)}, \quad r_{i,t}(\theta) = \frac{\pi_\theta(o_{i,t} \mid q, o_{i,<t})}{\pi_{\theta_{\text{old}}}(o_{i,t} \mid q, o_{i,<t})} \tag{14}$$

where $\hat{A}_{i,t}$ is an estimator of the advantage at time step $t$. Then, the objective function of GRPO is defined as follows:

$$
\mathcal{J}_{\text{GRPO}}(\theta) = \left[ \frac{1}{G} \sum_{i=1}^{G} \frac{1}{|o_i|} \sum_{t=1}^{|o_i|} \left( \min\left( r_{i,t}(\theta)\hat{A}_{i,t}, \ \text{clip}(r_{i,t}(\theta), 1-\epsilon, 1+\epsilon)\hat{A}_{i,t} \right) - \beta D_{\text{KL}}(\pi_\theta \| \pi_{\text{ref}}) \right) \right]
$$
(15)

where $\epsilon$ is the clipping range, $\beta$ controls the KL penalty, and $\pi_{\text{ref}}$ is the reference model.

## 3 EXPERIMENTS

### 3.1 EXPERIMENTAL SETTINGS

**Dataset and Metrics.** We evaluate model performance on four long-form factual benchmarks: FActScore (Min et al., 2023), Hallulens-LongWiki (abbreviated as LongWiki) (Bang et al., 2025), LongFact (Wei et al., 2024), and Factory-Hard (Chen et al., 2025a). Metrics include FActScore, Recall@K, Precision, F1@K and WR (Win Rate). Detailed descriptions of the training data processing pipeline are provided in Section 2.2 and the Appendix C.1. For a detailed description of the evaluation metrics and related configuration settings, please refer to Appendix C.4.

**Models and Baselines.** To comprehensively evaluate the effectiveness of our proposed method and ensure a fair and thorough comparison across different training paradigms and model scales, we conduct extensive experiments on Qwen2.5 family models of various sizes, including 7B, 14B, and 32B. Our baselines include a wide range of representative methods: pretrained base models (Base), prompting method CoVe (Dhuliawala et al., 2023), self-evaluation method Self-Eval-P(True) (Zhang et al., 2024b), supervised fine-tuned models (DeepSeek Distillation series (Guo et al., 2025)), unsupervised method Intuitor (Zhao et al., 2025), DPO (Rafailov et al., 2023) and GRPO (Shao et al., 2024) variants based on different reward signals (DPO + FActScore / KLC rewards and GRPO + FActScore / KLC rewards).

**Evaluation and Metrics.** We employ multiple metrics for a comprehensive assessment of long-form factuality: FActScore (Min et al., 2023) evaluates the overall factual precision of responses; Precision measures the accuracy of the facts presented by the model, calculated as Precision $= S/(S+N)$, where $S$ and $N$ are the numbers of supported and not-supported claims, respectively; Recall@K assesses the response's capacity to contain a sufficient amount of verifiable information, computed as Recall@K $= \min(S/K, 1)$; F1@K is the harmonic mean of precision and recall, given by F1@K $= 2 \cdot \text{Precision} \cdot \text{Recall@K}/(\text{Precision} + \text{Recall@K})$; additionally, we use the Win Rate (WR) based on judgments from GPT-4 to provide a comparative measure of the overall factuality and comprehensiveness of the models.

**Training Details.** Training is conducted on $16 \times$H100 GPUs (80GB) with full-parameter tuning. For all experiments, we employ a consistent learning rate of 1e-6 with cosine warmup. Further implementation details are provided in the Appendix C.2.

### 3.2 MAIN RESULTS

As summarized in Table 1, our method demonstrates significant improvements under both training paradigms. KLCF-zero, trained directly from the base model, outperforms all baselines, indicating effective factuality enhancement without SFT. Notably, it achieves a higher performance ceiling than training from the SFT model. We attribute this to the fact that initiating reinforcement learning directly from the base model mitigates the knowledge forgetting and the alignment tax incurred by SFT-based distillation (Luo et al., 2025; Fu et al., 2024). Compared to methods using FActScore-based rewards (like Self-Eval-P(True)), which heavily favor precision at the cost of recall, our framework jointly optimizes both objectives via dual-fact alignment, achieving the best overall balance and superior factual F1 scores. For detailed analysis of reward trends and training dynamics, please refer to Appendix D.2.

| Model | FActScore | LongWiki | | | LongFact | | | | Factory-Hard | | | |
|---|---|---|---|---|---|---|---|---|---|---|---|---|
| | FS | R@32 | Prec | F1@32 | R@64 | Prec | F1@64 | WR | R@64 | Prec | F1@64 | WR |
| **Training from Base Model** | | | | | | | | | | | | |
| Base (10-shot) | 46.8% | 0.661 | 0.435 | 0.511 | 0.614 | 0.685 | 0.637 | - | 0.239 | 0.302 | 0.262 | - |
| + CoVe | 46.0% | 0.435 | 0.381 | 0.368 | 0.380 | 0.676 | 0.452 | 19.8% | 0.179 | 0.304 | 0.206 | 17.9% |
| + Self-Eval-P(true) | 52.0% | 0.498 | **0.567** | 0.504 | 0.232 | **0.718** | 0.338 | 8.63% | 0.126 | 0.337 | 0.173 | 25.9% |
| + KLCF-zero (Ours) | **61.2%** | **0.681** | 0.552 | **0.568** | **0.776** | 0.704 | **0.733** | **94.6%** | **0.296** | **0.350** | **0.309** | **65.5%** |
| **Training from SFT Model** | | | | | | | | | | | | |
| SFT (Distill) | 48.7% | 0.503 | 0.532 | 0.494 | 0.403 | 0.760 | 0.507 | - | 0.138 | 0.286 | 0.175 | - |
| + Intuitor | 52.7% | 0.505 | 0.580 | 0.512 | 0.397 | 0.774 | 0.502 | 41.5% | 0.145 | 0.294 | 0.183 | 41.8% |
| + DPO&FActScore | 53.0% | 0.502 | 0.538 | 0.496 | 0.407 | 0.765 | 0.512 | 47.2% | 0.158 | 0.308 | 0.190 | 44.4% |
| + DPO&KLC | 54.9% | 0.505 | 0.584 | 0.517 | 0.432 | 0.779 | 0.530 | 56.9% | 0.157 | 0.312 | 0.186 | 52.4% |
| + GRPO&FActScore | **57.6%** | 0.472 | **0.647** | 0.510 | 0.406 | 0.780 | 0.504 | 29.6% | 0.143 | 0.351 | 0.184 | 30.2% |
| + KLCF (Ours) | 55.7% | **0.663** | 0.597 | **0.651** | 0.668 | 0.783 | 0.692 | 79.3% | 0.221 | 0.383 | 0.242 | 66.5% |

Table 1: Main results on Qwen2.5-14B. The upper and lower sections show results from training on the base model and the SFT model (DeepSeek-R1-Distill-Qwen-14B). "R@32" and "Prec" denote Recall@32 and Precision. Best scores are bolded.

| Reward Weight | | | FActScore | LongWiki | | | LongFact | | | | Factory-Hard | | | |
|---|---|---|---|---|---|---|---|---|---|---|---|---|---|---|
| Recall $\kappa$ | Precision $\lambda$ | Truth $\mu$ | FS | R@32 | Prec | F1@32 | R@64 | Prec | F1@64 | WR | R@64 | Prec | F1@64 | WR |
| 0 | 0 | 0 | 46.8% | 0.661 | 0.435 | 0.511 | 0.614 | 0.685 | 0.637 | - | 0.239 | 0.314 | 0.262 | - |
| $\frac{1}{3}$ | $\frac{2}{3}$ | 0 | 48.8% | 0.678 | 0.424 | 0.508 | 0.729 | 0.672 | 0.695 | 95.0% | 0.243 | 0.266 | 0.248 | **81.5%** |
| $\frac{1}{3}$ | $\frac{1}{3}$ | $\frac{1}{3}$ | 55.7% | **0.695** | 0.490 | 0.545 | 0.748 | 0.689 | 0.713 | **96.4%** | 0.273 | 0.258 | 0.290 | 79.3% |
| $\frac{1}{4}$ | $\frac{1}{4}$ | $\frac{1}{2}$ | 61.2% | 0.681 | 0.552 | **0.568** | 0.776 | 0.704 | 0.733 | 94.6% | **0.296** | 0.350 | **0.309** | 65.5% |
| 0.4 | 0.2 | 0.4 | 57.5% | 0.692 | 0.512 | 0.553 | **0.781** | 0.706 | **0.737** | 95.4% | 0.281 | 0.323 | 0.292 | 70.3% |
| 0 | $\frac{1}{3}$ | $\frac{2}{3}$ | 69.5% | 0.434 | 0.679 | 0.464 | 0.642 | 0.761 | 0.685 | 49.2% | 0.241 | 0.435 | 0.290 | 22.9% |
| $\frac{1}{3}$ | 0 | $\frac{2}{3}$ | 59.5% | 0.672 | 0.542 | 0.563 | 0.734 | 0.689 | 0.721 | 93.5% | 0.283 | 0.320 | 0.298 | 57.8% |
| 0 | 0 | 1 | **69.6%** | 0.369 | **0.724** | 0.422 | 0.633 | **0.793** | 0.685 | 36.8% | 0.243 | **0.492** | 0.296 | 15.3% |

Table 2: Ablation study on reward components. Results are for Qwen2.5-14B under the RL-zero (training from base model) setting. The first row is the base model (10-shot) reference, and the gray row denotes our optimal reward weighting.

## 3.3 ABLATION STUDY

As shown in Table 2, ablating the Dual-Fact Alignment mechanism validates its critical role. Using only the Checklist Reward ($\kappa = 1/3, \lambda = 2/3, \mu = 0$) yields high recall but low precision, as excessive coverage introduces errors. Conversely, only the Truthfulness Reward ($\kappa = 0, \lambda = 0, \mu = 1$) boosts precision at a severe cost to recall, indicating over-conservatism. Systematically adjusting the reward weights (from top to bottom in Table 2) reveals a clear trade-off: reducing the Checklist weight while increasing the Truthfulness weight monotonically improves precision but consistently degrades recall. This trend highlights the limitation of optimizing either objective in isolation. By integrating both rewards, KLCF achieves a superior balance. The Truthfulness Reward suppresses hallucinations encouraged by the Checklist Reward, which in turn counteracts conservatism, demonstrating the necessity of our dual mechanism.

## 3.4 SCALING STUDY

To verify the generalizability of the KLCF method across different model scales, we conduct scaling experiments on Qwen2.5-7B and 32B models. As shown in Table 3, for both the 7B and 32B models, our proposed method under the "Training from Base Model" setting significantly outperforms the original base model and the CoVe prompting approach across all factuality metrics. The results demonstrate that the performance improvement brought by KLCF is consistent regardless of model scale. Our method effectively enhances long-form factuality in small (7B), medium (14B), and large (32B) models, confirming that the framework possesses strong scalability and generalizability.

| Model | FActScore | LongWiki | | | LongFact | | | | Factory-Hard | | | |
|---|---|---|---|---|---|---|---|---|---|---|---|---|
| | FS | R@32 | Prec | F1@32 | R@64 | Prec | F1@64 | WR | R@64 | Prec | F1@64 | WR |
| **Qwen2.5-7B** | | | | | | | | | | | | |
| Base (10-shot) | 36.6% | 0.529 | 0.359 | 0.412 | 0.522 | 0.600 | 0.548 | - | 0.203 | 0.267 | 0.223 | - |
| + CoVe | 40.1% | 0.293 | 0.357 | 0.294 | 0.326 | 0.605 | 0.383 | 15.5% | 0.171 | 0.278 | 0.192 | 11.8% |
| + KLCF-zero (Ours) | **55.0%** | **0.633** | **0.487** | **0.514** | **0.724** | **0.675** | **0.693** | **94.8%** | **0.258** | **0.301** | **0.271** | **78.9%** |
| **Qwen2.5-32B** | | | | | | | | | | | | |
| Base (10-shot) | 40.0% | 0.664 | 0.453 | 0.520 | 0.531 | 0.691 | 0.575 | - | 0.234 | 0.318 | 0.256 | - |
| + CoVe | 45.9% | 0.375 | 0.410 | 0.355 | 0.379 | 0.718 | 0.466 | 19.5% | 0.191 | 0.346 | 0.229 | 15.3% |
| + KLCF-zero (Ours) | **59.3%** | **0.812** | **0.532** | **0.631** | **0.784** | **0.747** | **0.760** | **96.2%** | **0.307** | **0.434** | **0.322** | **71.3%** |

Table 3: Scaling study of KLCF on Qwen2.5-7B and 32B models under the "Training from Base Model" setting.

## 3.5 GENERALIZATION TO NON-THINKING MODELS

While our primary experiments demonstrate the effectiveness of KLCF starting from a base model under a thinking-style architecture, we further investigate its generalization capability by applying it to a standard non-thinking model. For this experiment, we adapt the reward function by removing the format reward (as it is unnecessary for a standard conversational model) and the general reward (relying instead on the KL divergence penalty to prevent excessive deviation), while retaining the length penalty to control verbosity. Results show that KLCF still achieves a substantial improvement in factuality metrics (as shown in Table 4), confirming that its effectiveness is not reliant on a thinking-style architecture and generalizes robustly to conventional conversational models.

| | FActScore | LongWiki | | | LongFact | | | | Factory-Hard | | | |
|---|---|---|---|---|---|---|---|---|---|---|---|---|
| | FS | R@32 | Prec | F1@32 | R@64 | Prec | F1@64 | WR | R@64 | Prec | F1@64 | WR |
| Qwen2.5-14B-Instruct | 61.1% | 0.548 | 0.548 | 0.578 | 0.412 | 0.788 | 0.531 | - | 0.193 | 0.388 | 0.245 | - |
| + KLCF (Ours) | **65.0%** | **0.708** | **0.580** | **0.595** | **0.686** | **0.808** | **0.747** | **97.6%** | **0.290** | **0.398** | **0.306** | **67.1%** |

Table 4: Performance improvement on factuality benchmarks after adapting KLCF to a non-thinking, instruction-tuned model (Qwen2.5-14B-Instruct).

## 3.6 TRUTHFULNESS REWARD ANALYSIS

We conducted a comparative analysis between the standard truthfulness reward calculation (Eq. 8) and its variant (Eq. 9) that leverages the pre-verified factual checklist to reduce evaluation noise. Both methods enhance factuality, but they exhibit distinct characteristics in balancing precision and recall. Due to space constraints, a detailed discussion of the results and their implications is provided in Appendix D.4.

## 3.7 EFFICIENCY ANALYSIS

To evaluate the efficiency of our reward design, we compare the computational cost of various factuality reward methods. Experimental details are provided in Appendix D.3. As summarized in Table 14, our combined reward (Checklist + Truthfulness) achieves a $3.46\times$ and $4.09\times$ speedup over FActScore and VeriScore in serial mode, which further increases to $5.24\times$ and $5.40\times$ under parallel execution. More importantly, our method eliminates external search engine APIs, using only lightweight, locally deployed reward models. This reduces token consumption and boosts inference speed, making it ideal for online reinforcement learning training.

## 3.8 HALLUCINATION TAX ANALYSIS

The "hallucination tax" is typically defined as the degradation in factuality that occurs when aligning models for other desirable traits, such as helpfulness. To validate the effectiveness of KLCF in mitigating this issue, we design an experiment to examine how factual performance changes when optimizing for a non-factuality objective. We initialize all experiments from the SFT model

(DeepSeek-R1-Distill-Qwen-14B). First, we establish a helpfulness-only RL model by performing reinforcement learning solely on 5,000 non-factual instructions from UltraFeedback[1], using the win rate against the SFT model's response, judge by Qwen2.5-72B-Instruct, as the reward signal to singularly optimize helpfulness. To evaluate KLCF, we then train the KLC&helpfulness RL model from the same SFT initialization, but on a mixed dataset containing an equal number of non-factual and factual instructions. For this mixed training, we apply the win-rate reward and our KLC reward to their respective subsets of non-factual and factual data.

| | FActScore | LongWiki | | | LongFact | | | | Factory-Hard | | | | Alpaca Eval 2.0 |
|---|---|---|---|---|---|---|---|---|---|---|---|---|---|
| | FS | R@32 | Prec | F1@32 | R@64 | Prec | F1@64 | WR | R@64 | Prec | F1@64 | WR | |
| SFT (Distill) | 48.7% | 0.503 | 0.532 | 0.494 | 0.403 | 0.760 | 0.507 | - | 0.138 | 0.286 | 0.175 | - | 24.4% |
| + helpfulness-only RL | 46.5% | 0.521 | 0.521 | 0.468 | 0.449 | 0.732 | 0.540 | 52.6% | 0.153 | 0.243 | 0.183 | 56.7% | 34.3% |
| + KLC&helpfulness RL | **51.8%** | **0.579** | **0.545** | **0.532** | **0.505** | **0.743** | **0.577** | **67.1%** | **0.180** | **0.322** | **0.238** | **63.6%** | **35.2%** |

Table 5: Hallucination Tax Analysis Results.

As shown in Table 5, the helpfulness-only RL model attains a substantially higher AlpacaEval-2.0 score than the SFT baseline (from 24.4% to 34.3%), but this comes at a clear cost: a noticeable drop in factual metrics like FActScore. This pattern clearly demonstrates the hallucination tax. In contrast, the KLC&helpfulness RL model effectively preserves the high helpfulness performance on AlpacaEval-2.0 benchmark while simultaneously recovering and improving factual scores. These results indicate that, when mixed with general-task RL, KLCF can eliminate the hallucination tax without sacrificing helpfulness and even further enhance factuality.

### 3.9 CASE STUDY

In this section, we present a case study to qualitatively illustrate the impact of our KLCF framework. We compare the responses of the Qwen2.5-14B-Instruct model and its KLCF-enhanced version to a specific query from the LongFact test set. The results are shown in Table 16. Our analysis shows that the KLCF-trained model produces a significantly longer and more comprehensive response, indicating a substantial improvement in factual recall by utilizing more of its internal knowledge. Furthermore, the enhanced output also demonstrates higher factual precision, with fewer observable errors compared to the base model. This dual improvement highlights the effectiveness of our approach in mitigating hallucinations while reducing over-conservatism. For a detailed analysis, including the specific query and the model outputs with annotated errors, please refer to Appendix D.5.

## 4 CONCLUSION, LIMITATIONS, AND FUTURE WORK

This paper proposes KLCF, a novel RL framework that mitigates hallucinations by explicitly aligning a policy model's expressed knowledge with its pretrained parametric knowledge to achieve knowledge-level consistency. Through its Dual-Fact Alignment mechanism, KLCF jointly optimizes factual recall and precision to balance comprehensiveness with truthfulness. Extensive experiments show it significantly outperforms existing baselines. Moreover, the efficient reward design enables scalable online RL training without costly external knowledge retrieval.

**Limitations and Future Work.** Our approach operates in a closed-book QA setting, which inherently restricts the model to its internal knowledge and excludes externally retrieved information. Moreover, KLCF currently applies factuality rewards at the response level, lacking fine-grained supervision over the intermediate reasoning process.

In the future, we plan to explore step-wise factual alignment, such as applying GRPO at the sentence level or introducing process-based rewards to better guide the model's chain of thought. It is also worth noting that while both the fact checklist construction and evaluation in this work rely on a static knowledge source (Wiki20250716), the general framework of KLCF is not limited by the knowledge source and can be easily extended to incorporate real-time search engines or human-annotated data.

---

[1]https://huggingface.co/datasets/HuggingFaceH4/ultrafeedback_binarized

## ETHICS STATEMENT

This work presents a reinforcement learning framework aimed at improving the factuality of large language models (LLMs) in long-form generation, with the primary goal of mitigating harmful hallucinations and enhancing the reliability of AI-generated content. Our research utilizes publicly available datasets (e.g., ELI5) or synthetically generated data based on publicly available sources (e.g., Wikipedia), which do not contain private or sensitive personal information. We acknowledge that while our work seeks to reduce the generation of incorrect information, the potential for LLMs to be misused for generating convincing but false content remains a broader societal challenge. We have conducted our research with a commitment to the Code of Ethics, focusing on a beneficial application of AI. To the best of our knowledge, this work does not present any immediate, direct, or novel risks to privacy, security, or fairness beyond those common to the field of LLM development. We fully adhere to the Code of Ethics and affirm that there are no potential conflicts of interest to disclose.

## REPRODUCIBILITY STATEMENT

To facilitate the reproducibility of our work, we have released the complete source code, including our reinforcement learning framework and reward model implementations, on an anonymous GitHub repository: `https://anonymous.4open.science/r/KLCF-DE13`. We provide detailed descriptions of our data processing pipeline, encompassing knowledge elicitation, claim extraction, and verification, in Appendix C.1. Furthermore, all experimental hyperparameters and configuration details for training and evaluation are comprehensively documented in Appendix C.2 and Appendix C.4. The base models used in our studies are publicly available pretrained checkpoints. We have endeavored to ensure that the core contributions of KLCF—its algorithm and reward design—can be accurately reproduced based on the descriptions in this paper and the provided resources.

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

APPENDICES

## A  STATEMENT ON LLM USAGE

This paper employed a Large Language Model (GPT-5 by OpenAI) in a limited capacity, strictly as a writing aid. The model's role was confined to proofreading and polishing select passages to enhance readability and language fluency. Specifically, it was used to rephrase sentences for better flow and correct minor grammatical errors in introductory, explanatory, and concluding sections. Crucially, the LLM did not contribute to the core research process. It was not used for generating ideas, designing experiments, conducting analyses, or interpreting results. The fundamental scientific contributions and the intellectual substance of the paper are solely the product of the authors' work. The LLM served only as a passive tool to assist in the final presentation of the manuscript.

## B  RELATED WORK

### B.1  LONG-FORM FACTUALITY EVALUATION

Fine-grained factuality assessment of generated text serves as a fundamental prerequisite for reliable factuality alignment. Early approaches typically decompose long-form text into atomic facts and verify them against external knowledge to evaluate factual accuracy. FActScore (Min et al., 2023)

and FacTool (Chern et al., 2023) establish this paradigm, employing retrieval and large verification models to determine the veracity of each atomic fact. Subsequently, methods such as SAFE (Wei et al., 2024) and VeriScore (Song et al., 2024) further refine the fact extraction and verification pipeline, while numerous follow-up studies (Ni et al., 2024; Bishop et al., 2024; Wanner et al., 2025) continue to advance this direction by improving evaluation effectiveness and efficiency. However, these methods generally rely on external retrieval interfaces and large-scale verification models, resulting in high computational overhead and making them unsuitable for training scenarios in online reinforcement learning that require high-frequency reward signals.

## B.2 LONG-FORM FACTUALITY ALIGNMENT

To directly enhance the factuality of LLMs, researchers have proposed various alignment methods. Among approaches based on supervised fine-tuning (SFT) and direct preference optimization (DPO), methods such as FactTune-FS (Tian et al., 2023) and FLAME (Lin et al., 2024a) leverage scores from evaluators like FActScore to guide models toward factual preferences, while FactAlign (Huang & Chen, 2024) and Mask-DPO (Gu et al., 2025) construct preference pairs based on external retrieval for DPO training. However, these methods are generally confined to offline training scenarios and often prioritize factual precision at the expense of recall, leading to overly conservative model behavior.

In recent years, a growing body of work has begun incorporating factuality signals as rewards within RL frameworks. For example, KnowRL (Ren et al., 2025) introduces factuality rewards based on knowledge verification during training to help the model recognize its knowledge boundaries, and TruthRL (Wei et al., 2025) proposes a triple reward mechanism that distinguishes between correct answers, hallucinations, and abstentions, encouraging the model to refrain from responding when uncertain to improve truthfulness. Additionally, several other studies (Chen et al., 2025b; Zhang et al., 2025a; Li & Ng, 2025) explore the use of various types of factuality signals as rewards to enhance model capability and reduce hallucinations. Nevertheless, most of these methods still rely on time-consuming external retrieval processes, which limits their applicability in large-scale online RL training. In contrast to the above works, the proposed KLCF framework designs a fully external-knowledge-free and lightweight reward mechanism. Through dual-fact alignment, KLCF jointly optimizes factual recall and precision, effectively enhancing factuality while avoiding the decline in expressiveness caused by excessive conservatism.

## B.3 RL WITHOUT EXTERNAL FEEDBACK

To overcome the limitations of verifiers, researchers are increasingly turning to reward signal generation methods that operate without external feedback. These label-free RL approaches primarily develop along two representative directions. One line of research focuses on deriving reward signals from models' internal confidence estimates, enhancing predictive certainty by rewarding low-entropy or self-consistent outputs (Zhao et al., 2025; Zhang et al., 2025c; Shafayat et al., 2025; Agarwal et al., 2025; Li et al., 2025; Prabhudesai et al., 2025). For instance, Intuitor (Zhao et al., 2025) leverages the model's own confidence estimations as unsupervised optimization signals to effectively improve the factuality of generated content. The other predominant paradigm constructs supervisory signals through collective decision-making mechanisms (Zhang et al., 2025d; Zuo et al., 2025), as exemplified by TTRL (Zuo et al., 2025), which aggregates majority-voted answers from multiple generated samples to serve as pseudo-ground-truth labels for reinforcement learning updates.

## B.4 ALIGNMENT TAX

The alignment tax is a prevalent phenomenon in RLHF and SFT processes. Some studies (Ouyang et al., 2022; Huang et al., 2025b; Lin et al., 2024b) indicate that alignment often leads models to become "overly conservative", sacrificing capability for improved safety. Some models like the DeepSeek-R1-Distill series perform well on general instruction-following tasks, their performance on long-form factuality benchmarks often falls short of the original base models, serving as a concrete manifestation of the alignment tax (See Table 1). More critically, current FActScore-guided RL methods further exacerbate this issue: to avoid negative judgments from external verifiers, models adopt a "less-is-better" generation strategy, leading to a substantial decline in recall and ultimately

reducing the overall F1 score. Therefore, a key motivation of this work is to mitigate the alignment tax by conducting RL directly from the base model with knowledge-level consistency as the objective. This approach enables the model to fully express its parametric knowledge while avoiding content that exceeds its knowledge boundaries.

# C    IMPLEMENTATION DETAILS

## C.1    DATA CONSTRUCTION

**RL Training Dataset.** ELI5 (Fan et al., 2019) is a publicly available long-form question-answering dataset, from which we select 7993 samples for training. To avoid overlap with evaluation benchmarks, we construct two new datasets adhering to established methodologies: LongFact-Gen, containing 4348 samples generated by reproducing the original LongFact (Wei et al., 2024) prompt strategy using GPT-4.1; and LongWiki-Gen, comprising 2248 samples built from the GoodWiki (Choi, 2023) corpus following the procedure described in HalluLens (Bang et al., 2025). Both training sets are distinct from their corresponding test benchmarks.

We then follow the data preprocessing pipeline outlined in Section 2.2 to perform uniform cleaning and formatting on all three datasets. The training datasets are summarized in the table 6.

| | RL | | | DPO |
|---|---|---|---|---|
| | **Dataset Size** | **Avg. True Claims** | **Avg. Checklists** | **Dataset Size** |
| 7B | 12680 | 34.76 | 10.09 | 12680 |
| 14B | 13230 | 38.34 | 11.03 | 13230 |
| 32B | 12967 | 37.19 | 11.10 | 12967 |

Table 6: Statistics of training data.

**DPO Training Dataset.** In this work, we employ the DeepSeek-R1-Distill-Qwen model for DPO training. To construct the DPO training dataset, we first sample 8 responses for each query from the RL dataset using three different sizes of models (7B, 14B and 32B). Subsequently, we extract claims from all responses for every query using a lightweight trained claim extraction model along with Prompt 2. We then verify all claims of each response against a locally built Wiki20250716 index, utilizing the Qwen2.5-72B-Instruct and Prompt 3, to obtain the FActScore for every response. Finally, we form the <chosen, rejected> pairs by selecting the highest and lowest-scoring responses for each query, thereby completing the preparation of the DPO dataset.

Similarly, to construct DPO training data based on the reward mechanism proposed in this paper, we need to assign a reward value to each response generated for every query, according to the reward computation method defined in Section 2.3. Specifically, we first use a lightweight checklist verification model with Prompt 7 to extract the checklist verification results for each response, and then compute the reward $R_{\text{recall}}$ and the reward $R_{\text{precision}}$ based on Eq. (5) and Eq. (6), respectively. Subsequently, based on the previously extracted claims, we employ a truthfulness reward model with Prompt 6 to obtain the credibility probability score for each claim, and calculate the reward $R_{\text{truth}}$ according to Eq. (8). Then, the final reward value for each response is computed using Eq. (13). Finally, for each query, we select the responses with the highest and lowest reward values to form the <chosen, rejected> preference pairs. The training datasets used for DPO are summarized in the table 6.

**Test Dataset.** In this paper, we conduct a comprehensive evaluation of the trained models on four public long-form text evaluation benchmarks: FActScore (Min et al., 2023), Hallulens-LongWiki (Bang et al., 2025), LongFact (Wei et al., 2024), and Factory (Chen et al., 2025a). The statistics for these datasets are detailed in Table 7. Specifically, for FActScore, we use 500 samples provided in its official release as the test set. For Hallulens-LongWiki, we strictly follow the methodology described in the original paper to generate 250 test samples. For the LongFact benchmark, we select 250 samples from its "Objects" subset. For the Factory benchmark, we randomly select 250 samples from its "Hard" subset to form the test set.

|  | FActScore | Hallulens-LongWiki | LongFact-Objects | Factory-Hard |
|---|---|---|---|---|
| Total Dataset Size | 500 | - | 1140 | 421 |
| Test Dataset Size | 500 | 250 | 250 | 250 |

Table 7: Statistics of test dataset.

## C.2 TRAINING SETUPS

In this section, we systematically elaborate on the hyperparameter configurations employed across all core experiments in this paper, covering training stages such as Supervised Fine-Tuning (SFT), Direct Preference Optimization (DPO), and Reinforcement Learning (RL). To ensure the reproducibility and consistency of our experiments, hyperparameter settings are kept uniform within the same training stage. For the comparative model Intuitor, we adopt the relevant configurations from its original paper. All specific parameter values are summarized in Table 8.

## C.3 BASELINES

This section provides a detailed description of the baseline methods included in our experiments to ensure a comprehensive and fair comparison. Our evaluation encompasses a diverse set of representative approaches across different training paradigms.

The **Base** model refers to the pretrained Qwen2.5 models (7B, 14B, 32B) evaluated in a 10-shot setting, serving as the foundational performance benchmark. The prompting-based baseline is represented by **CoVe** (Dhuliawala et al., 2023), which reduces hallucinations through a self-verification mechanism during inference without updating model parameters. For supervised fine-tuning, we include the **DeepSeek Distillation Series** (Guo et al., 2025). We also compare against Intuitor (Zhao et al., 2025), an unsupervised method that improves factuality by leveraging the model's intrinsic confidence estimates.

Furthermore, we incorporate several reinforcement learning baselines to isolate the contribution of our reward design. This includes DPO and GRPO optimized with different reward signals: **DPO + FActScore** and **GRPO + FActScore** use the external FActScore metric as the reward signal, favoring precision but often at the cost of recall. In contrast, **DPO + KLC Rewards** and **GRPO + KLC Rewards** (which is our full KLCF method) are trained using the knowledge-level consistency rewards introduced in this work, enabling joint optimization of recall and precision through dual-fact alignment.

## C.4 EVALUATION SETUPS

### C.4.1 SETUPS

In this study, we adopt corresponding metrics for different evaluation benchmarks. For FActScore (Min et al., 2023), we directly use the factscore metric proposed in the original paper, calculating the scores for each model based on its officially released 500 test data points. For the 250 samples from Hallulens-LongWiki (Bang et al., 2025), we likewise employ the three metrics recommended by the original paper: Recall@32, Precision, and F1@32. For the LongFact and Factory datasets, we adopt the evaluation pipeline proposed in VeriScore (Song et al., 2024) and report three key metrics: Recall@64, Precision, and F1@64. Additionally, for the LongFact and Factory datasets, we introduce the Win Rate (WR) metric to provide a complementary performance perspective.

**FActScore.** The automated FActScore evaluation in the original paper's code[2] typically relies on models such as ChatGPT for atomic fact decomposition and uses its provided outdated Wiki knowledge base for judgment. However, due to the high cost of ChatGPT and the outdated nature of the original Wiki knowledge base, we make adjustments to this evaluation pipeline.

To address these issues, our approach adopts a more cost-effective and timely solution. Specifically, we use the powerful open-source model Qwen2.5-72B-Instruct to replace ChatGPT for decompos-

---

[2]https://github.com/shmsw25/FActScore

**RL and DPO**

| task | framework | data | actor_rollout_ref | algorithm | trainer |
|---|---|---|---|---|---|
| RL | VeRL | max_prompt_length=8192
max_response_length=4096
train_batch_size=64 | entropy_coeff=0.0
use_kl_loss=False
kl_loss_coef=0.0
rollout_n=8
ppo_epochs=1
ppo_mini_batch_size=64
lr=1e-6,weight_decay=0.1
lr_warmup_steps_ratio=0.1
warmup_style=cosine
temperature=1.0, top_p=0.05
gpu_memory_utilization=0.5
tensor_model_parallel_size=4
gradient_checkpointing=True | gamma=1.0
lam=1.0
$L_m$=2048
$L_m - L_c$=850 | total_epochs=2
nnodes=2
n_gpus_per_node=8
save_freq=20 |
| Intuitor | VeRL | max_prompt_length=8192
max_response_length=4096
train_batch_size=64 | use_kl_loss=True
kl_loss_coef=0.05 | same as above | same as above |
| DPO | LLaMA Factory | cutoff_len=4096
val_size=0.05
template=qwen | per_device_train_batch_size=1
gradient_accumulation_steps=8
per_device_eval_batch_size=1
lr_scheduler_type=cosine
warmup_ratio=0.1 | pref_beta=0.1
pref_loss=sigmoid | num_train_epochs=2
logging_steps=1
save_steps=96
eval_strategy=steps |

**SFT**

| task | framework | data | model | optim | trainer |
|---|---|---|---|---|---|
| Checklit Verifier
Claim Extractor | VeRL | train_batch_size=64
micro_batch_size_per_gpu=1
max_length=12288
truncation=right | model_dtype=fp32
cpu_offload=False
offload_params=False
gradient_checkpointing=True | lr=1e-6
betas=[0.9, 0.95]
weight_decay=0.01
warmup_steps_ratio=0.1
clip_grad=1.0
lr_scheduler=cosine | total_epochs=3
nnodes=1
n_gpus_per_node=8 |
| Truthfulness
Reward Model | VeRL | max_length=4096 | same as above | lr=1e-5 | Same as above |

Table 8: A systematic summary of training hyperparameter settings to ensure reproducibility.

ing text into atomic facts. Concurrently, we build a local Wiki20250716 index to serve as the latest knowledge base. Based on the information retrieved from this knowledge base, we also use the Qwen2.5-72B-Instruct model to judge the veracity of each atomic fact. This pipeline ensures the evaluation's cost-effectiveness, data timeliness, and result reliability.

**Hallulens-LongWiki.** In evaluating on the Hallulens-LongWiki[3] benchmark, we follow its standard factuality framework of claim extraction, retrieval, and verification. While we use the benchmark's official knowledge base for retrieval, we employ Qwen2.5-72B-Instruct for the extraction and verification steps to ensure a cost-effective and reproducible setup.

**LongFact.** For the LongFact[4] benchmark, we randomly sample 250 instances from its more challenging "Objects" subset to form the test set. The SAFE framework proposed in LongFact is a complex, high-cost advanced evaluation framework that relies on Google Search. To enable quick and efficient evaluation, we instead use VeriScore to assess the LongFact dataset in this paper. VeriScore is an improved version of FActScore and SAFE, which optimizes the process by focusing on extracting and verifying "verifiable claims" and introducing inter-sentence context to improve extraction quality, thereby avoiding unnecessary revision steps. In this paper, we consistently use the powerful Qwen2.5-72B-Instruct to complete all steps of the VeriScore evaluation.

**Factory.** Factory[5] is a large-scale, human-verified, and challenging prompt set. We randomly select 250 samples from its "Hard" subset for evaluation. Similar to LongFact, we compute all three metrics using the VeriScore pipeline with a local Wiki knowledge and the Qwen2.5-72B-Instruct model.

To ensure the reproducibility of our experiments and the transparency of the evaluation pipeline, we have compiled a detailed summary of the key parameter configurations used throughout the process. These configurations cover every step from fact decomposition to final verification. The specific parameter settings are detailed in Table 9.

| Claim Extraction | Wiki Retrieval | Claim Verification |
|---|---|---|
| temperature=0.1
max_tokens=8192 | top_k=10
chunk_size=300
chunk_overlap=20 | temperature=0.1
max_tokens=8192 |

Table 9: The key parameter configurations used in the evaluation pipeline.

### C.4.2 METRICS

This section provides formal definitions of the evaluation metrics used in our study to assess the long-form factuality of model responses. The reported values for Precision, Recall@K, and F1@K in our experiments are the average of each metric calculated for every individual response in the test set.

**Precision.** Precision measures the factual accuracy of a generated response. It is defined as the proportion of individual facts within the response that are verifiable as supported against an external knowledge source (e.g., web search results). Let $S(y)$ be the number of supported facts in a response $y$, and $N(y)$ be the number of not-supported facts. Precision for a single response is calculated as:

$$\text{Precision}(y) = \frac{S(y)}{S(y) + N(y)} \tag{16}$$

A higher precision indicates that the responses contain fewer factual errors on average.

**Recall@K.** Recall evaluates the comprehensiveness of a response. In open-domain long-text generation, defining the complete set of expected facts is infeasible. Following prior work (Wei et al., 2024), we adopt a parametric recall metric. Let $K$ be a hyperparameter representing a user's desired number of supported facts for a high-quality response. Recall@K for a single response is calculated as:

$$\text{Recall@K}(y) = \min\left(\frac{S(y)}{K}, 1\right) \tag{17}$$

This metric measures whether a response provides an adequate amount of verifiable information, up to a specified limit $K$.

---

[3] https://github.com/facebookresearch/HalluLens
[4] https://github.com/google-deepmind/long-form-factuality
[5] https://huggingface.co/datasets/facebook/FACTORY

**F1@K.** To balance the trade-off between factual precision and informational comprehensiveness (Recall@K), we use their harmonic mean to compute the F1@K score for a single response:

$$F_1@K(y) = \begin{cases} \frac{2 \cdot \text{Precision}(y) \cdot \text{Recall@K}(y)}{\text{Precision}(y) + \text{Recall@K}(y)}, & \text{if } S(y) > 0 \\ 0, & \text{if } S(y) = 0 \end{cases} \tag{18}$$

A response achieves a high F1@K score only by being both highly factual and sufficiently detailed.

**WR.** Win Rate (WR) is a comparative metric designed to evaluate the performance of a candidate model (B) relative to a baseline model (A). For each prompt in the test set, responses from both models are evaluated by a judge LLM (GPT-4.1 using Prompt 8 (Li et al., 2023)), which receives two outputs and assigns a ranking based on factuality and comprehensiveness. To control for position bias, each pairwise comparison is performed twice under reversed output ordering. In the first trial, Model A's response is provided as output1 and Model B's as output2; we record whether B is ranked higher than A, denoted as $\text{Win}^{(1)}$. In the second trial, the order is reversed: Model B's response is given as output1 and Model A's as output2, producing a result $\text{Win}^{(2)}$. The win indicator for the $i$-th instance is the average of both trials:

$$\text{Win} = \frac{\text{Win}^{(1)} + \text{Win}^{(2)}}{2} \tag{19}$$

The overall WR of model B over A on a test set of size $N$ is defined as:

$$\text{WR}_{B \text{ vs } A} = \frac{\text{Win}}{N} \tag{20}$$

A WR greater than 0.5 indicates that the candidate model (B) is preferred over the baseline (A). Unlike other metrics, WR provides a single aggregate measure of relative performance over the entire test set.

## D  ADDITIONAL EXPERIMENTS

In this section, we provide additional experimental details not covered in the main body, aiming to supplement and expand upon our findings. This content primarily includes supplementary analysis of our core experiments and other results not discussed in detail in the main text.

### D.1  VERIFIER TRAINING

**Claim Extraction Model Training.** To build an efficient and lightweight factual claim extraction model, we adopt the following training pipeline. First, we leverage the Claimify method proposed in (Metropolitansky & Larson, 2025), which has a significant advantage in generating structured factual claims. We use Qwen2.5-72B-Instruct as the data construction engine, utilizing a carefully designed prompt template to obtain verifiable claims. This process ultimately results in a large-scale, high-quality training dataset containing 10187 entries. Subsequently, we use Qwen2.5-14B-Instruct as the base model and perform supervised fine-tuning on this self-constructed dataset, yielding our final lightweight claim extraction model. This model maintains high-precision extraction capabilities while significantly reducing computational overhead, allowing it to be seamlessly integrated into subsequent online reinforcement learning loops.

As shown in Table 10, we conducted a case study to demonstrate the practical efficacy of the trained claim extraction model. The model successfully decomposed a complex long-form response into a set of independent atomic facts. This case shows that the model strictly adheres to the three principles emphasized in the prompt: Atomicity, Objective Verifiability, and Explicit Meaning. It accurately decomposed a response containing multiple facts into independent, complete, and unambiguous assertions, providing perfect input for the subsequent verification step.

**Checklist Verifier Training.** To ensure the discriminative capability and generalizability of the Checklist Verifier, we train the model using a high-quality dataset of 10139 samples accumulated from previous research. First, we use the DeepSeek-R1-0528 to generate entirely new and diverse

**Query:**

Can you provide a detailed overview of Marie Curie's life and scientific contributions?

**Response:**

Marie Curie was born on November 7, 1867, in Warsaw, Poland. She was a pioneering physicist and chemist who conducted groundbreaking research on radioactivity. Curie was the first woman to win a Nobel Prize and the only person to win Nobel Prizes in two different scientific fields—Physics in 1903 and Chemistry in 1911. She discovered the elements polonium and radium, which revolutionized the field of nuclear physics. Curie's work has had a lasting impact on modern medicine, particularly in cancer treatment. Despite facing significant gender barriers, she is widely remembered as a brilliant and determined scientist.

**Claims:**

* Marie Curie was born on November 7, 1867
* Marie Curie was born in Warsaw, Poland
* Marie Curie was a pioneering physicist
* Marie Curie was a pioneering chemist
* Marie Curie conducted groundbreaking research on radioactivity
* Marie Curie was the first woman to win a Nobel Prize
* Marie Curie is the only person to have won Nobel Prizes in two different scientific fields
* Marie Curie won the Nobel Prize in Physics in 1903
* Marie Curie won the Nobel Prize in Chemistry in 1911
* Marie Curie discovered the element polonium
* Marie Curie discovered the element radium
* Marie Curie's discoveries of polonium and radium revolutionized the field of nuclear physics
* Marie Curie's work has had a lasting impact on modern medicine
* Marie Curie's work has had a lasting impact on cancer treatment

Table 10: A case study demonstrating the atomic claim extraction from a model's response.

responses for each training prompt. This step effectively enhances the model's robustness in judging texts of varying styles. Subsequently, we perform post-processing on these generated responses by removing any chain-of-thought content. This directs the model's focus toward factual verification rather than mimicking reasoning processes. Finally, we conduct supervised fine-tuning on the processed dataset using Qwen2.5-14B-Instruct as the base model, resulting in an efficient and lightweight verifier model.

We evaluate the SFT model on two self-constructed test sets: an English set with 81 questions and a Chinese set with 147 questions. The detailed evaluation results are presented in the Table 11.

|        | English-81Q | Chinese-147Q |
|--------|-------------|--------------|
| epoch1 | 69/81=0.852 | 128/147=0.871 |
| epoch2 | **73/81=0.901** | **129/147=0.878** |
| epoch3 | 70/81=0.864 | 129/147=0.878 |

Table 11: Based on the evaluation of accuracy on both the English (81Q) and Chinese (147Q) test sets, we select the checkpoint from epoch 2 as the final Checklist Verifier model.

**Truthfulness Reward Model Training.** The training of the Truthfulness Reward Model constitutes a crucial component of our framework. Specifically, after verifying all claims using Qwen2.5-72B-Instruct and the Wiki20250716 knowledge base, we obtain the verification results as summarized in Table 12, where "SUPPORT" and "REFUTE" represent positive and negative samples, respectively.

|      | ELI5 | | | LongFact-Gen | | | LongWiki-Gen | | |
|------|---------|--------|-----------------|---------|--------|-----------------|---------|--------|-----------------|
|      | SUPPORT | REFUTE | NOT ENOUGH INFO | SUPPORT | REFUTE | NOT ENOUGH INFO | SUPPORT | REFUTE | NOT ENOUGH INFO |
| 7B   | 298822  | 14758  | 57376           | 149420  | 12854  | 46928           | 50260   | 13408  | 38990           |
| 14B  | 319512  | 9220   | 44304           | 168417  | 7757   | 36744           | 63920   | 9415   | 33760           |
| 32B  | 308659  | 7254   | 39929           | 163364  | 6903   | 33607           | 60035   | 9980   | 32779           |

Table 12: Distribution of claim verification results based on Qwen2.5-72B-Instruct and Wiki20250716.

Given the significant scarcity of negative samples compared to positive ones, we perform the following sampling strategy: all negative samples are duplicated three times, while positive samples are down-sampled to achieve a positive-to-negative ratio of 2:1. Through this process and based on Prompt 6, we construct training datasets suitable for models of three different scales: 7B, 14B, and 32B. We then conduct SFT on pre-trained models of these three sizes to obtain reward models capable of assessing claim truthfulness based on the model's inherent knowledge boundaries.

|  | Model | Accuracy | F1 |
|---|---|---|---|
| 7B (3000Q) | Qwen2.5-7B | 0.7237 | 0.7684 |
|  | Base-few-shot | 0.7813 | 0.7741 |
|  | SFT | **0.8187** | **0.8272** |
| 14B (3000Q) | Qwen2.5-14B | 0.6450 | 0.6321 |
|  | Base-few-shot | 0.8096 | 0.8119 |
|  | SFT | **0.8347** | **0.8421** |
| 32B (3000Q) | Qwen2.5-32B | 0.7777 | 0.7752 |
|  | Base-few-shot | 0.8200 | 0.8210 |
|  | SFT | **0.8334** | **0.8423** |

Table 13: Performance comparison of Truthfulness Reward Models at different scales on the test sets. The results show that models after SFT significantly outperform both the base model and the base few-shot model in terms of Accuracy and F1-score.

After model training is completed, we perform a systematic evaluation of its performance on independently constructed test sets. For the three model sizes—7B, 14B, and 32B—we construct separate test sets, each containing 3000 cases with a positive-to-negative sample ratio of 1:1. The test results of the models are detailed in Table 13. The table shows that the models after SFT exhibit significant improvements in both Accuracy and F1-score compared to the base model and the base few-shot model.

## D.2 MAIN RESULTS

In this section, we present a more detailed analysis of the training process and model performance through various curves that illustrate the dynamics of our KLCF framework, as summarized in Fig. 4.

**Knowledge-level Consistency Rewards.** All factual rewards show a significant upward trend, indicating that KLCF-zero training effectively optimizes the model's core objectives. Among them, the `Fact Recall` reward increases markedly from approximately 0.4 to around 0.75, demonstrating the greatest improvement. This confirms that the factual coverage (comprehensiveness) of the model's generated responses improves substantially, as the model learns to recall and output more facts within its knowledge boundaries. The `Fact Precision` reward rises further from a relatively high starting point (0.88) to near 0.98, indicating that the model maintains extremely high factual accuracy (correctness) while becoming more comprehensive. The `Truthfulness` reward remains at a very high level throughout ($0.85 \rightarrow 0.95$), reflecting the base model's inherent strong truthfulness foundation, which our method further enhances.

**Auxiliary Rewards.** Changes in the auxiliary rewards reflect optimization of the model's overall quality. The `General Reward` (from the Skywork-Reward model) steadily increases from 0.1 to 0.8, indicating significant improvement in the overall quality, fluency, and human preference alignment of the generated responses. The `Format Reward` converges stably from an initial penalty of approximately -0.6 to 0, indicating that the model initially made formatting errors but quickly learned and fully mastered the required output structure (<think>...</think> and <answer>...</answer>), eventually incurring no further penalty. The `Length Penalty` fluctuates slightly near 0 throughout, demonstrating that the generation length remains effectively constrained within the predefined optimal range ($L_m - L_c = 850$ tokens), avoiding redundant or overly short responses.

**KL Divergence.** The KL divergence increases slowly from 0 and stabilizes at around 0.005. This slight yet consistent increase is expected and indicates that the policy model undergoes a controlled deviation from the original base model policy in order to optimize the factual rewards.

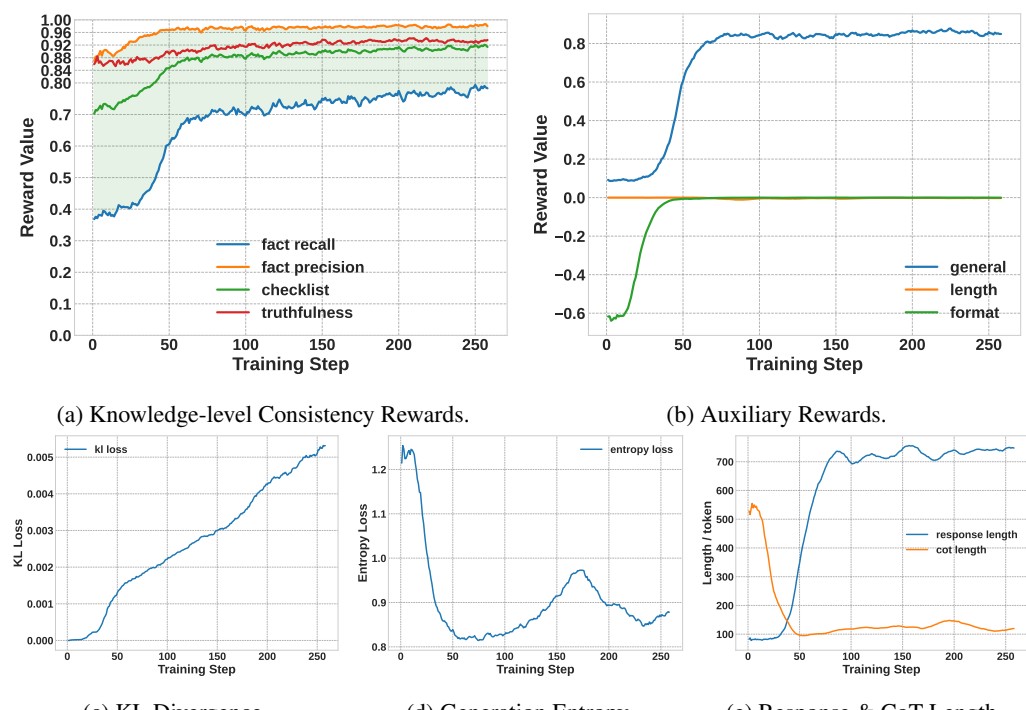

(a) Knowledge-level Consistency Rewards.

(b) Auxiliary Rewards.

(c) KL Divergence.

(d) Generation Entropy.

(e) Response & CoT Length.

Figure 4: Training Dynamics of KLCF-zero on Qwen2.5-14B. The figure illustrates the progression of key metrics throughout the reinforcement learning process. (a) The core knowledge-level consistency rewards, all showing significant improvement. (b) The auxiliary rewards guiding response quality and structure. (c) The KL divergence, measuring the deviation from the base model. (d) The entropy of actor model's generation probabilities, reflecting the policy's exploration-exploitation balance. (e) The lengths of the generated responses and the internal reasoning chains.

**Generation Entropy.** The actor model's generation entropy decreases rapidly in the early stages of training and then stabilizes. This trend indicates that the model quickly sharpens its policy by reducing the randomness of its choices to obtain higher rewards. The subsequent stabilization suggests that the model reaches a relatively stable optimization plateau, achieving a good balance between exploration and exploitation.

**Response Length and CoT Length.** The total Response Length starts at around 100 tokens, gradually increases as training progresses, and eventually stabilizes around 700 tokens. This change directly correlates with the rise in the Fact Recall reward, as the model naturally generates longer responses to cover more factual points. Its eventual stabilization indicates effective constraint by the Length Penalty.

D.3    EFFICIENCY ANALYSIS

This section provides additional implementation details for the efficiency analysis presented in Section 3.7, ensuring the fairness, reproducibility, and credibility of our results.

**Data and Model Configuration.** To guarantee a broad and unbiased evaluation, we randomly select 50 prompts from the training set of the Qwen2.5-14B model to form the evaluation dataset. To completely eliminate the influence of the model's training state on the length and structure of the generated content, we use an untrained DeepSeek-R1-Distill-Qwen-14B to generate responses for all prompts. These responses serve as the unified input for all reward calculation methods.

**Our Reward Calculation Setup.** For the calculation of our proposed rewards, the scale of the reward models and verifiers remains consistent with the main experiments. Specifically, both

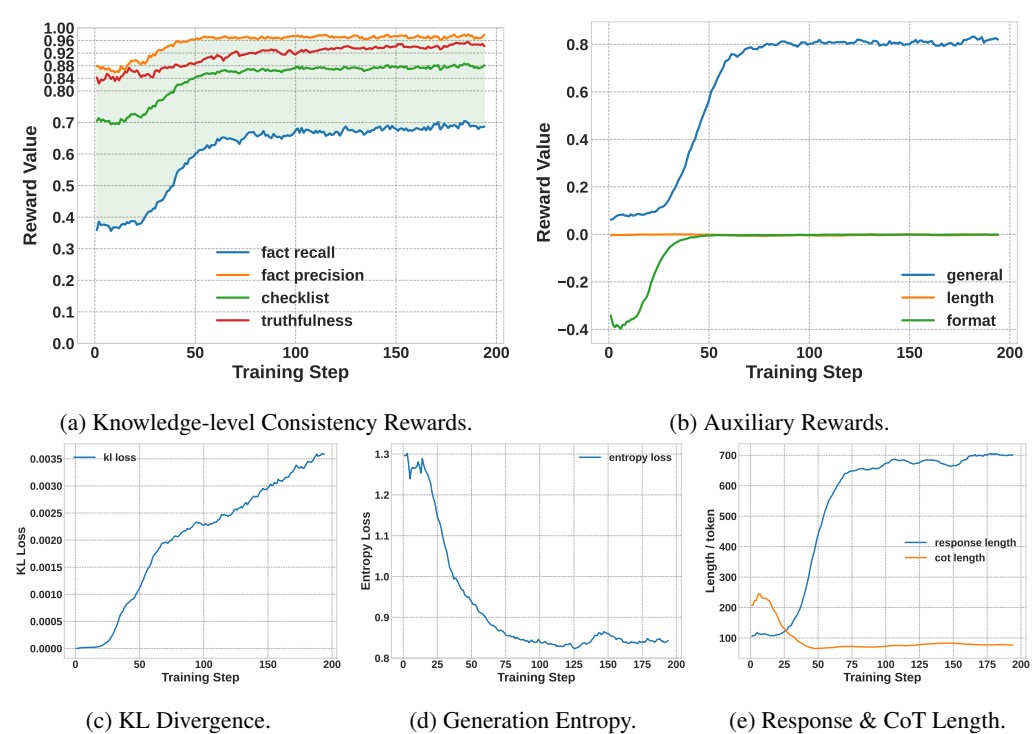

(a) Knowledge-level Consistency Rewards.

(b) Auxiliary Rewards.

(c) KL Divergence.

(d) Generation Entropy.

(e) Response & CoT Length.

Figure 5: Training Dynamics of KLCF-zero on Qwen2.5-7B.

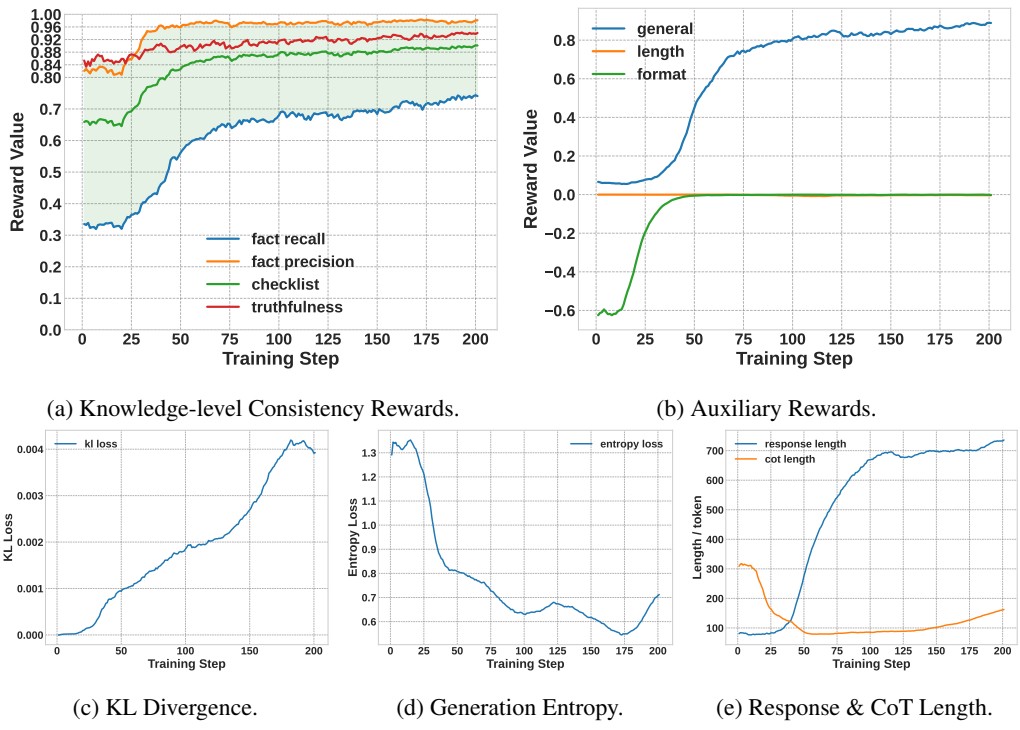

(a) Knowledge-level Consistency Rewards.

(b) Auxiliary Rewards.

(c) KL Divergence.

(d) Generation Entropy.

(e) Response & CoT Length.

Figure 6: Training Dynamics of KLCF-zero on Qwen2.5-32B.

the Checklist Verifier and the Truthfulness Reward Model are fine-tuned from the Qwen2.5-14B-Instruct, ensuring that the efficiency evaluation aligns with the actual application scenario.

| | Reward | Time (s) | ↑(vs. FActScore) | ↑(vs. VeriScore) | # Searches | Input Tokens | Output Tokens |
|---|---|---|---|---|---|---|---|
| Serial | FActScore | 198.06 | 1.00x | 1.18x | 51.74 | 55666.20 | 5701.56 |
| | VeriScore | 234.50 | 0.84x | 1.00x | 32.64 | 335909.78 | 3442.96 |
| | Ours | 57.29 | **3.46x** | **4.09x** | 0 | 1457.76 | 855.24 |
| Parallel | FActScore | 10.58 | 1.00x | 1.03x | 52.02 | 56677.50 | 5926.04 |
| | VeriScore | 10.90 | 0.97x | 1.00x | 32.56 | 334077.92 | 3407.28 |
| | Ours | 2.02 | **5.24x** | **5.40x** | 0 | 1452.62 | 851.46 |

Table 14: Efficiency comparison of different factual reward calculation methods. All reported values are averaged over 50 samples. Our proposed reward (Checklist + Truthfulness) achieves significant speedups in both serial and parallel modes. The "# Searches" column indicates the number of times the method called the external search engine API. Notably, our method requires no such calls and maintains low input & output token consumption.

| Reward | FActScore | LongWiki | | | LongFact | | | | Factory-Hard | | | |
|---|---|---|---|---|---|---|---|---|---|---|---|---|
| | FS | R@32 | Prec | F1@32 | R@64 | Prec | F1@64 | WR | R@64 | Prec | F1@64 | WR |
| Base (10-shot) | 46.8% | 0.661 | 0.435 | 0.511 | 0.614 | 0.685 | 0.637 | - | 0.239 | 0.302 | 0.262 | - |
| $R_{\text{truth}}$ | **61.2%** | **0.681** | **0.552** | **0.568** | **0.776** | **0.704** | **0.733** | **94.6%** | **0.296** | 0.348 | **0.309** | **65.5%** |
| $R_{\text{truth}}^{\text{variant}}$ | 60.7% | 0.643 | 0.548 | 0.545 | 0.751 | 0.691 | 0.714 | 91.0% | 0.284 | **0.350** | 0.299 | 56.2% |

Table 15: Comparison between the standard truthfulness reward and its variant. Experiments were conducted by performing RL from the base Qwen2.5-14B model.

**Baseline Methods Setup.** To ensure a fair comparison, we standardize the internal components of the baseline methods, FActScore and VeriScore. We replace their original claim extraction and claim verification modules with the same-sized Qwen2.5-14B-Instruct. This step eliminates the potential impact of model scale differences on computational efficiency, focusing the comparison solely on the overhead introduced by the algorithmic designs.

**Deployment and Computational Environment.** We deploy all models using the vLLM inference framework on two H100 GPUs. When evaluating the parallel computing performance of all methods, we set the concurrency to 100 to effectively simulate the high-throughput reward calculation scenario common in large-scale reinforcement learning training.

### D.4 TRUTHFULNESS REWARD ANALYSIS

This section provides a comparative analysis of the standard Truthfulness Reward (Eq. 8) and its variant (Eq. 9), under the RL-zero training setting on Qwen2.5-14B. While both reward formulations significantly improve factuality over the base model, the variant exhibits a slight but consistent performance drop across most metrics compared to the standard version.

The observed performance gap can be primarily attributed to the inherent limitations of the factual checklist upon which the variant relies. Although the checklist is constructed from verified facts within the base model's knowledge boundary, it is not exhaustive. Omissions or errors during the checklist construction pipeline—such as incomplete claim extraction or imperfect verification—lead to an incomplete prior. Consequently, the variant reward may incorrectly evaluate certain valid claims as "new" and subject them to the noisy estimation of the reward model, rather than leveraging them as high-confidence signals. This introduces inconsistency in the reward signal during policy optimization.

Furthermore, while the variant was designed to reduce evaluation noise by focusing only on claims outside the checklist, this very mechanism restricts its supervisory scope. The standard reward, by evaluating every claim in the response, provides a more comprehensive and stable learning signal. The variant's narrower focus appears to slightly hinder the policy's ability to generalize toward a balanced factuality profile, particularly in recall-oriented scenarios. Thus, despite its conceptual appeal, the variant's dependency on a potentially imperfect checklist diminishes its effectiveness relative to the broader and more robust standard reward.

## D.5 CASE STUDY

In this section, we conduct a qualitative study on the performance of the KLCF framework on specific samples. Specifically, we compare Qwen2.5-14B-Instruct and its KLCF-trained version on an example from the LongFact test set. The results are shown in Table 16.

The analysis shows that the output of the model after KLCF training is significantly longer and more detailed. This directly reflects a substantial improvement in **factual recall**, indicating that the model more actively and completely utilizes the information in its pre-trained knowledge base, rather than adopting a conservative strategy. This improvement primarily stems from our proposed Factual Checklist Reward mechanism, which incentivizes the model to cover a broader range of known facts during generation, thereby effectively mitigating over-conservatism.

More importantly, while the output content is expanded, its **factual precision** is also simultaneously improved. The output of the model without KLCF training contains some vague descriptions and information gaps, whereas the output of the KLCF model not only includes more factual details but is also described more accurately and specifically, without introducing observable new errors. This validates the effectiveness of the Dual-Fact Alignment mechanism: the Truthfulness Reward, based on the model's internal knowledge for self-assessment, guides the model to generate claims with higher confidence, thus effectively reducing the production of inaccurate or fabricated content.

In summary, this case qualitatively demonstrates that KLCF enables the model to achieve a better balance between factual recall and precision.

# E PROMPTS

This section provides all prompts employed in this study. The **red bold** portions within the prompts indicate placeholders to be filled.

## E.1 DATA CONSTRUCTION

---

**Prompt for Base Model Generation**

Please answer the following questions in a given format.
[Question]: query example 1
[Answer]: response example 1

[Question]: query example 2
[Answer]: response example 2
...
[Question]: query example 10
[Answer]: response example 10

[Question]: **{query}**
[Answer]:

---

**Qwen2.5-14B-Instruct Response:**

Amal Clooney is a renowned international human rights lawyer and advocate based in London and New York. She is known for her work on high-profile cases involving human rights violations, freedom of expression, and other significant legal issues around the world. Amal has represented individuals and organizations in various countries, including Lebanon, Egypt, and Azerbaijan, often focusing on cases that involve press freedom and the rights of journalists.

**She was born on April 18, 1978, in Beirut, Lebanon**, and **grew up there before moving to England at age 13**. Amal attended **St. Paul's Girls' School** and later studied law at **St Edmund Hall**, Oxford University, and **Middle Temple**, where she was called to the bar in 2002. She went on to complete an LLM at **Columbia Law School in New York City**.

In addition to her legal work, Amal Clooney is also recognized for her philanthropic efforts and advocacy for various causes. She is married to the actor George Clooney, with whom she shares two children. Together, they have been involved in several charitable initiatives and campaigns aimed at promoting human rights and social justice.

**+ KLCF Response:**

Amal Clooney is a renowned international human rights lawyer and advocate known for her work on high-profile cases and her commitment to humanitarian issues around the world. Born on February 3, 1978, in Beirut, Lebanon, she has gained significant public attention not only due to her professional achievements but also because of her marriage to the actor George Clooney, though her career and legal work stand out as her primary contributions.

### Key Aspects of Amal Clooney's Career

1. **Education and Early Career**: Amal Clooney earned her undergraduate law degree in Jurisprudence at St Hugh's College, Oxford, and later obtained an LLM from New York University School of Law. Her academic background laid a strong foundation for her legal career, focusing on international law and human rights advocacy.

2. **Legal Expertise**: She specializes in international law, particularly in areas such as freedom of expression, press freedom, and human rights violations. She has represented clients in cases before the International Criminal Court (ICC), European Court of Human Rights, and other international tribunals.

3. **High-Profile Cases**: Clooney has been involved in several notable cases that have garnered global media attention. These include defending journalists and activists against charges that could be seen as politically motivated or unjust, such as the case of the Al Jazeera journalists detained in Egypt, and advocating for victims of war crimes and human rights abuses.

4. **Humanitarian Work**: Beyond her legal practice, she works with various organizations to promote human rights and supports initiatives aimed at protecting journalists and whistleblowers. She has been vocal about issues related to refugees and has worked on campaigns to raise awareness and improve conditions for those seeking asylum.

5. **Public Advocacy and Speeches**: She often speaks at international forums, conferences, and universities...

6. **Philanthropy and Social Causes**: Clooney is associated with multiple charities and foundations that focus on education...

7. **Professional Networks and Collaborations**: She collaborates with international legal bodies, non-governmental organizations...

8. **Personal Life and Public Image**: While her personal life, including her marriage to George Clooney...

### Impact and Legacy

Amal Clooney's impact extends beyond individual cases; she contributes to shaping legal strategies and international norms regarding human rights and justice. Her work helps set precedents that can influence how governments and international bodies handle human rights issues and prosecute crimes against humanity.

Her approach to law combines rigorous legal analysis with a strategic use of media and public opinion to achieve justice for her clients and to push for broader systemic changes. This dual approach—combining traditional legal advocacy with modern communication strategies—has made her one of the most influential figures in international human rights law.

### Challenges and Criticisms

Despite her accolades, Clooney faces criticism from some quarters regarding the effectiveness of her methods, the balance between her public persona and substantive legal outcomes, and occasional controversies surrounding the high-profile nature of her cases versus more systemic issues faced by less visible groups.

Overall, Amal Clooney is recognized for her dedication to advancing human rights and justice on a global scale, using her skills and platform to address complex legal and ethical challenges facing our interconnected world. Her work continues to inspire and influence legal practitioners and human rights advocates worldwide.

Table 16: Responses to the query *"Who is lawyer Amal Clooney?"* from the Qwen2.5-14B-Instruct model and its KLCF-enhanced counterpart. Text highlighted in **red bold** across both responses indicates factually inaccurate claims. The KLCF-generated response is notably longer and more comprehensive, encompassing a greater number of verifiable factual details while containing fewer overall factual errors.

---

**Prompt for Claim Extraction**

Below you will receive a piece of text. Your task is:

1. Determine whether the text contains verifiable objective claims.
2. If verifiable objective claims exist in the text, you must extract these claims from the answer (regardless of whether these claims are true).
3. If the text does not contain any verifiable objective claims, return "no verifiable objective claims".

Response format:

* Claim 1
* Claim 2
...
(or "no verifiable objective claims")

The claims you extract must adhere to the following 3 principles:

1. Objectively verifiable: The claim must describe an objectively verifiable fact, not a subjective judgment, evaluation, or opinion.
2. Indivisible: The objective fact described by the claim cannot be further broken down.
3. Explicit meaning: Each claim must be a complete, self-contained sentence with all coreferences resolved. There should be no nouns or pronouns with unclear meaning.

Please strictly follow the above rules to complete the following task:
[Text]: **{response}**
[Verifiable objective claims]:

---

---

**Prompt for Claim Verification**

You will be provided with a [CLAIM] and several pieces of reference [EVIDENCE]. Your task is to determine the relationship between the facts in the [CLAIM] and the [EVIDENCE].

You need to analyze each piece of [EVIDENCE] in relation to the given [CLAIM] and then provide an overall conclusion.

- **SUPPORT**: The facts in the [CLAIM] appear in the [EVIDENCE] and are consistent with the descriptions. Or, the facts in the [CLAIM] can be correctly inferred from the information in the [EVIDENCE].
- **REFUTE**: The facts in the [CLAIM] contradict the information in the [EVIDENCE]. Or, the facts in the [CLAIM] can be inferred to be false based on the information in the [EVIDENCE].
- **NOT ENOUGH INFO**: The facts in the [CLAIM] are neither supported by nor contradicted by the information in the [EVIDENCE], or a definitive conclusion cannot be drawn from the [EVIDENCE].

Your response must be a dictionary with a single key-value pair, where the key is "conclusion" and the value is the overall conclusion (**SUPPORT**, **REFUTE**, or **NOT ENOUGH INFO**). You must strictly follow the format below, returning a dictionary in JSON format. Do not return any other content.

[RESPONSE FORMAT]:
```json
{
    "conclusion": "The overall conclusion, return one of **SUPPORT**, **REFUTE**, or **NOT ENOUGH INFO**"
}
```

Now, complete the following task.
[CLAIM]: {claim}

[EVIDENCE]:
{evidence}

Your decision:

---

---

**Prompt for Claim Prioritization**

You are given a user query and a list of candidate claims.

Your task:
- Retain only the claims that are essential and highly relevant to the query.
- Eliminate duplicates. If two claims capture slightly different aspects, preserve both.
- Each claim must convey exactly one idea, expressed as a clear, explicit, and self-contained sentence.
- If conflicting claims exist, present them in the form: "Some evidence suggests [...], while others indicate [...]".
- Output format:
\boxed{
    key claim 1
    key claim 2
    key claim 3
    ...
}
- Provide output strictly within \boxed{}.

Output Format:
\boxed{
    Final Key Claim 1
    Final Key Claim 2
    ...
}

Input:
- Query: **{query}**
- Candidate Claims:
**{claims}**

Output:

---

## E.2 TRAINING

---

**Prompt for RL-zero**

A conversation between User and Assistant. The user asks a question, and the Assistant solves it. The assistant first thinks about the reasoning process in the mind and then provides the user with the answer. The reasoning process and answer are enclosed within <think> </think> and <answer> </answer> tags, respectively, i.e., <think> reasoning process here </think> <answer> answer here </answer>. User: **{prompt}** Assistant: <think>

---

**Prompt for Truthfulness Reward Model Training**

You need to evaluate the factual correctness of the following claim and output either "True" (the claim is entirely factually correct) or "False" (the claim contains errors or goes beyond your knowledge).

Claim: **{claim}**
Factual correctness:

---

---

Prompt for Checklist Verification

You will receive a **Question**, a **Reply** and a **Fact List**. Your task is to check, one by one, how each fact in the list is covered in the reply and decide whether the fact is **Consistent**, **Contradictory**, or **Missing**.

**Input**

* [Question]: a factual question.
* [Reply]: the text answer that needs to be evaluated.
* [Fact List]: a list of factual points to verify.

**Output**

Return a list in **strict JSON format**. For every fact in the list, output a dictionary containing:

* "analysis": a brief analysis of how the reply aligns with this fact.
* "conclusion": one of "Consistent", "Contradictory", or "Missing", indicating how the fact is covered in the reply.

* **"Consistent"**: the core information of the fact appears in the reply and matches the description.
* **"Contradictory"**: some information in the fact conflicts with information in the reply; from this fact you can infer that part of the reply is incorrect.
* **"Missing"**: the fact is neither fully implied nor contradicted by the reply; the reply does not mention the fact or omits some core information.

Now, using the **Question**, **Reply**, and **Fact List** given below, analyze each fact and output the results strictly in the required format:

[Question]: **{query}**
[Reply]: **{response}**
[Fact List]:
**{guidelines}**

[Output]:

---

## E.3 EVALUATION

---

**Prompt for Calculating Win Rate**

```
<|im_start|>system
```
You are a helpful assistant, that ranks models by the quality of their answers.
```
<|im_end|>
<|im_start|>user
```
I want you to create a leaderboard of different of large-language models. To do so, I will give you the instructions (prompts) given to the models, and the responses of two models. Please rank the models based on which responses would be preferred by humans. All inputs and outputs should be python dictionaries.

Here is the prompt:
{
    "instruction": """**{instruction}**"""
}

Here are the outputs of the models:
[
    {
        "model": "model_1",
        "answer": """**{output_1}**"""
    },
    {
        "model": "model_2",
        "answer": """**{output_2}**"""
    }
]

Now please rank the models by the quality of their answers, so that the model with rank 1 has the best output. Then return a list of the model names and ranks, i.e., produce the following output:
[
    {'model': <model-name>, 'rank': <model-rank>},
    {'model': <model-name>, 'rank': <model-rank>}
]

Your response must be a valid Python dictionary and should contain nothing else because we will directly execute it in Python. Please provide the ranking that the majority of humans would give.
```
<|im_end|>
```

---

