# OpenReview forum: "Knowledge-Level Consistency Reinforcement Learning: Dual-Fact Alignment for Long-Form Factuality"
_ICLR.cc/2026/Conference — Submitted to ICLR 2026_

### Official Review · Reviewer_7TbV · 2025-10-27

**Soundness:** 3
**Presentation:** 2
**Contribution:** 2
**Rating:** 4
**Confidence:** 5

**Summary:**

This paper proposes KLCF to address hallucinations in large language models' long-form generation. KLCF aligns the model's expressed knowledge with its internal parametric knowledge using a Dual-Fact Alignment mechanism that jointly optimizes factual recall and precision. KLCF uses lightweight, external-knowledge-free rewards, enabling efficient online training. Experiments across multiple benchmarks and model scales show factuality improvements while maintaining scalability.

**Strengths:**

1. This paper implements an online RL training algorithm for hallucination mitigation that achieves alignment between the model's internal knowledge and its knowledge utilization strategy.
2. The reward function design effectively balances factual precision and recall, addressing a critical trade-off in hallucination mitigation. As demonstrated in Table 2, different objective emphases lead to opposing model tendencies, revealing that precision and recall are inherently conflicting optimization targets.
3. The method is validated across different model sizes and both thinking/non-thinking architectures, with comprehensive evaluation metrics.

**Weaknesses:**

1. The claim of being "external-knowledge-free" is inappropriate since both the truthfulness reward and checklist rely on offline Wikipedia verification. The offline and online phases should not be treated separately when making this claim.

2. While claiming to be "lightweight," the offline data preparation and reward model training costs should not be ignored. Moreover, the time ratio between response rollout and reward computation during GRPO training should be provided to assess the significance of the efficiency gains from reward computation improvements.

3. The reward system appears overly complex, with Truthfulness Reward and fact precision reward seemingly optimizing similar objectives. I have concerns about the necessity of Checklist Reward. A simpler approach using shaped rewards combining FActScore (precision) + number of claims (recall) could achieve similar alignment between parametric and expressed knowledge (e.g., https://arxiv.org/abs/2406.12221). The authors should better explain the rationale for eliciting knowledge from pretraining.

4. The paper should include a Related Work section discussing the relationship and distinctions with other hallucination mitigation approaches.

**Questions:**

1. Training Dynamics and Reward Scaling: Section C.2 shows CoT becomes shorter while response length increases during training. This seems more attributable to General Reward (which prefers longer responses) rather than recall reward. Although the coefficient is 0.1 in Formula 10, could the General Reward have a much larger numerical scale? Could the authors provide the actual value ranges of different reward components to clarify their relative impacts?

2. How are rewards computed in practice? Is it using separately deployed GPUs as in Table 12? If all 16 training GPUs are used for reward scoring instead, would the efficiency gains become much less significant?

---

> ### Author Response · Authors · 2025-11-21
> **Authors' Rebuttal 1**
>
> We sincerely thank the reviewer for their positive feedback on the effectiveness and generalizability of our method, as well as the careful analysis of the precision-recall trade-off. We have carefully considered the weaknesses and questions raised, and will provide detailed responses regarding the use of external knowledge, computational efficiency, reward design rationale, related work discussion, training dynamics, and reward scaling in the revised manuscript.
>
> **W1**: **Clarification on the "external-knowledge-free" claim**
>
> Thank you for pointing out the lack of precision in our "external-knowledge-free" claim; we fully agree with your observation.
>
> The core innovation of our method lies in achieving online reinforcement learning training without the need for real-time external knowledge calls. Unlike methods such as FActScore and SAFE, which require retrieving and verifying external knowledge in each training iteration, we only use Wikipedia during the offline preparation phase to construct static checklists and reward model training data. Once these lightweight resources are prepared, the subsequent large-scale online RL training is entirely independent of external knowledge sources, resulting in significant efficiency gains and scalability advantages (as shown in Table 14).
>
> Following your suggestion, we will revise the wording throughout the paper to more accurately describe our contribution as "online external-knowledge-free reinforcement learning." We sincerely appreciate your valuable feedback, which has helped us communicate the core features of our method more clearly.
>
> **W2**: **Efficiency Analysis**
>
> We thank the reviewer for raising the question regarding computational efficiency. A comprehensive analysis addressing this point has been provided in our **General Response 1 to Common Reviewers' Comments** under the section "**Q1: Is our KLCF framework truly efficient?**" We hope the detailed breakdown and discussion there will fully address your concern.
>
> **W3**: **Justification for the dual-reward complexity versus simpler precision-recall reward alternatives**
>
> We thank the reviewer for their insightful comments, which have prompted us to clarify the core rationale behind our reward design more clearly.
>
> 1.  **Regarding the Complementarity of Reward Components**
>     The Checklist-based Fact Precision Reward and the Truthfulness Reward are distinct in both objective and mechanism. The **Checklist-based Fact Precision Reward** provides precise supervision targeting a **known subset of knowledge**, ensuring the model accurately expresses verified internal knowledge. In contrast, the **Truthfulness Reward** acts as a **generalized truthfulness supervisor**, performing probabilistic assessments on all claims in the generated content, primarily suppressing fabrication in unknown domains. Since the checklist is prepared and verified offline (potentially offering higher precision than the online Truthfulness Reward), we intended for the Checklist-based Fact Precision Reward to further strengthen the reward signal for factual accuracy. However, our ablation study (Table 2) indeed shows that, under the current experimental setup, removing the Checklist-based Fact Precision Reward while retaining the Truthfulness Reward has a limited impact on overall performance.
>
> 2.  **Regarding the Core Rationale for the Checklist and Comparison with Baselines**
>     We have compared our method against the "FActScore + number of claims" scheme on the 14B model. Specifically, we refer to the scheme in [1], where the reward is defined as **FActScore + 0.1 * log(1 + *p*)**, with *p* being the number of claims. As shown in the table below, this scheme achieves lower F1 scores than KLCF across several test sets. We posit that while the "FActScore + number of claims" approach is structurally simple and intuitive, its reliance on external retrieval introduces noise and struggles with claims that cannot be retrieved or verified precisely. This constrains the accuracy of the reward signal. Furthermore, this method rewards all claims equally regardless of importance, potentially encouraging the model to engage in reward hacking by generating redundant, correct but irrelevant statements. The performance comparison between KLCF and "FActScore + number of claims" also indirectly validates that the policy model's parametric knowledge does not exceed the base model's knowledge boundary during RL training, confirming the effectiveness of using the base model's parametric knowledge boundary for providing rewards.

---

> > ### Comment · Reviewer_7TbV · 2025-11-28
> >
> > I acknowledge that the authors have addressed my concerns W1 and W4 through paper revisions.
> >
> > Regarding the "lightweight" efficiency claims, I believe a more comprehensive discussion is needed. In my experience, computational time is significantly correlated with the length and detail of model responses (specifically, the number of extracted statements). The authors should provide this relevant information. Additionally, the authors mention that FActScore faces limitations from "network latency and API rate limits." I suspect this refers to their implementation using OpenAI API calls. However, I question whether this comparison setup is entirely fair. Today's open-source models are sufficiently capable of performing FActScore tasks, and the authors could consider deploying models locally for inference to ensure a more equitable comparison. Details about FactScore setting are missing.
> >
> > Concerning the end-to-end training time comparison, I maintain that offline preparation time should still be factored into the overall efficiency evaluation, though I acknowledge that the authors' method offers greater convenience for hyperparameter tuning. Moreover, I believe that online RL based on FActScore is not an ideal baseline for efficiency comparison, as, to my knowledge, this approach has not been adopted in practice. One reason may be that FActScore was originally designed for evaluation purposes, and its prompting strategy tends to decompose statements overly granularly, which is inherently unfriendly to training. I suggest that [https://arxiv.org/pdf/2311.08401] would serve as a more appropriate and practical baseline. This work represents one of the earliest training-based approaches to hallucination mitigation, utilizing FActScore for DPO training.
> >
> > Overall, if the authors consider efficiency to be a significant advantage of their algorithm, I recommend dedicating a separate section in the appendix to discuss computational efficiency in detail, providing comprehensive comparisons and analysis.

---

> > > ### Author Response · Authors · 2025-12-03
> > > **Authors' Response**
> > >
> > > Thank you for the additional guidance. Your observations are very insightful and have helped us clarify the key points that need refinement in our efficiency comparison. We will incorporate these into the manuscript to ensure a more rigorous presentation.
> > >
> > > 1.  **Regarding the Fairness of the FActScore Comparison**: We fully agree with the principle of fair comparison. In our experiments, the entire FActScore evaluation pipeline (including fact decomposition and verification) was performed using locally deployed Qwen2.5-72B-Instruct models and a local Wiki knowledge base, without calling any external paid APIs. The mention of "network latency and API rate limits" in the main text was intended to illustrate that reliance on external services could become a bottleneck in **practical industrial deployment**, not to imply an unfair experimental setup on our part. The longer training time for FActScore primarily stems from its need to input lengthy retrieved document contexts into the large model for claim-by-claim verification, which incurs significant computational overhead.
> > >
> > > 2.  **Regarding the Completeness of Baselines**: The DPO work based on FActScore that you recommended is indeed an important baseline. In fact, in the "Training from SFT Model" section of our main experiments (Table 1), we already included both **"DPO+FActScore"** and **"GRPO+FActScore"** as baselines. The results show that, on the same data, our method achieves superior performance on comprehensive metrics (such as F1 and Win Rate). We will more prominently cite and discuss this work in the revised manuscript and include it in our efficiency analysis for comparison.
> > >
> > > 3.  **Regarding Comprehensive Efficiency Analysis**: We accept your suggestion. In addition to supplementing the correlation analysis between response length, number of claims, and computation time, we will **establish a dedicated "Computational Efficiency Analysis" section in the paper appendix**. This section will systematically present end-to-end time (including offline preparation), detailed cost breakdowns for each stage, and fair comparisons with various methods, including the baseline you recommended. This will ensure our argument for efficiency advantages is more comprehensive and transparent.
> > >
> > > Once again, thank you for helping us enhance the rigor and clarity of our work.

---

> > ### Comment · Reviewer_7TbV · 2025-11-28
> >
> > **Regarding the Complementarity of Reward Components:** I believe that the "fact precision reward and truthfulness reward" are essentially consistent because, if only one of these rewards were adopted, both would naturally lead the model to adopt the strategy of "responding to knowledge the model is certain about; refusing to answer for uncertain knowledge," thereby minimizing regret. Both share a consistent assumption that the model can judge its own "certainty" about knowledge. However, they differ in terms of the knowledge domains they align with. The former aligns with the pretraining model's own knowledge (assuming it can be honestly probed through prompts and remains consistent with the downstream model), while the latter aligns with external knowledge sources. I would expect these two approaches to potentially achieve similar effects, and I do not consider the latter to be less scalable. After all, the latter can perform real-time fact decomposition and search-API calls for any real-time input and model response, while the former relies on pre-prepared probing of the pretraining model. I recommend that the authors provide a more detailed discussion in Section 3.3 and find better ways to present Table 2. The table should better demonstrate the direct relationships between Precision/Recall and various parameters, as well as the balance between them. Additionally, an experimental group with only Recall and Truth rewards with Precision Weight set to 0 is missing, which would demonstrate that their offline preparation is necessary.

---

> > > ### Author Response · Authors · 2025-12-03
> > > **Authors' Response**
> > >
> > > Thank you for raising this profound point once more, which has prompted us to further clarify the design rationale and necessity of the dual-reward mechanism.
> > >
> > > Regarding your discussion on **the complementarity of the reward components**, we have supplemented a more detailed analysis in Section 3.3 and updated the ablation studies. Key additions include:
> > >
> > > 1.  **On the Necessity of the `Fact Precision Reward`**: As suggested, we have added the experimental group **`(κ=1/3, λ=0, μ=2/3)`** (i.e., using only the `fact recall` and `truthfulness` rewards). Results show an observable decrease in **Precision** across all test sets compared to the full KLCF (e.g., a drop of approximately 2-3 percentage points on FActScore), though the magnitude of this decrease is smaller than when the `truthfulness` reward is completely removed. This indicates that the `fact precision reward`, based on the offline checklist, plays a distinct role in **anchoring and precisely expressing the core knowledge mastered by the model**. It forms an effective functional complement with the `truthfulness` reward, which is primarily responsible for suppressing hallucinations in unknown domains.
> > >
> > > 2.  **On Scalability and Method Positioning**: We fully agree that methods relying on external retrieval offer generality in knowledge coverage. The core objective of this work is precisely to address the **efficiency bottleneck** faced by such methods in **large-scale online RL training**—the substantial overhead caused by real-time retrieval and verification. KLCF is positioned as **efficient "closed-book" alignment**. By converting the cost of knowledge probing into a one-time offline investment, it achieves efficiency and stability in online training, which is key to scalability for RL training requiring numerous iterations. The two paradigms are, in fact, complementary solutions tailored for different scenarios.
> > >
> > > The relevant analysis and supplementary experiments have been integrated into the revised manuscript. We sincerely appreciate this highly insightful discussion, which has significantly enhanced the rigor and clarity of our work.

---

> ### Author Response · Authors · 2025-11-21
> **Authors' Rebuttal 2**
>
> | model | FActScore | LongWiki |  |  | LongFact |  |  |  | Factory-Hard |  |  |  |
> | :--- | :--- | :--- | :--- | :--- | :--- | :--- | :--- | :--- | :--- | :--- | :--- | :--- |
> | | FS| R@32 | Prec | F1@32 |  R@64 | Prec | F1@64 | WR | R@64 | Prec | F1@64 | WR | WR|
> |FActScore + num claims| 53.9% | 0.616 | 0.568 | 0.610 | 0.652 | 0.757 | 0.685 | 70.3% | 0.213 | 0.361 | **0.247** |61.5%
> | KLCF | **55.7%**  | **0.663** | **0.597** | **0.651** | **0.668** | **0.783** | **0.692** | **79.3%** | **0.221** | **0.383** | 0.242 | **66.5%**
>
> 3.  **Regarding the Necessity of Eliciting Knowledge from the Pretrained Model**
>     The theoretical foundation of our approach rests on the premise that, for a closed-book model, there exists no universal, complete external "set of facts" that can serve as the recall target. By eliciting knowledge from the base model itself to construct the checklist, we obtain a personalized recall benchmark that is **precisely aligned with the specific parametric knowledge boundary of that model**. This ensures the reward signal is always grounded in knowledge the model already possesses, forming the cornerstone for safe and precise knowledge alignment.
>
> **W4**: **Related work is missing.**
>
> Thank you for this suggestion. We agree that a dedicated discussion of related work is important.
>
> In the initial submission, we integrated discussions of prior art into the Introduction due to space considerations. Following your feedback, we have now added a comprehensive **Related Work** section as **Appendix B**.
>
> This new section systematically reviews four key areas:
> 1.  Long-Form Factuality Evaluation
> 2.  Long-Form Factuality Alignment
> 3.  RL Without External Feedback
> 4.  Alignment Tax
>
> In each part, we explicitly outline the connections to and key distinctions from existing methods, thereby clarifying our paper's specific contributions within the broader landscape of hallucination mitigation.
>
> We believe this addition significantly strengthens the paper and are grateful for the recommendation.
>
> **Q1**: **Explanation of reward scaling and the actual impact of the General Reward on output length**
>
> We thank the reviewer for this insightful observation. We agree that analyzing the actual impact of each reward component is crucial and offer the following clarifications:
>
> 1.  **The numerical range of the General Reward is limited:** We applied a Sigmoid function to the output of the reward model (Skywork-Reward), strictly confining its raw output to the [0, 1] interval. After applying the 0.1 coefficient, its **maximum contribution to the final reward is only 0.1**. Compared to core reward components like `R_recall` and `R_precision` (which typically range between 0.4-0.9), its actual influence is very limited and is certainly not the primary driver behind the increased answer length.
>
> 2.  **Increased answer length is driven by the recall reward and constrained by the length penalty:** The increase in answer length is primarily driven by the **recall reward**, which directly incentivizes the model to cover more facts from the checklist within the `<answer></answer>` section. Concurrently, our designed **length penalty** effectively prevents answers from growing indefinitely, ensuring the conciseness and information density of the generated content.
>
> 3.  **Regarding the shorter CoT:** With the exception of the format reward, all core factual rewards are computed solely based on the final answer and do not apply any optimization objective to the CoT itself. For general tasks like long-form QA (as opposed to complex mathematical reasoning), a longer preceding reasoning process does not necessarily lead to a better or more accurate final answer [2], and an ill-structured CoT can even increase the likelihood of hallucinations [3]. Furthermore, we observed that this phenomenon varies under different initial conditions (Base or SFT). In our experiments, when starting RL directly from the Base model, the model learned to use more concise reasoning chains to organize its thoughts during training, thereby allocating more expressive capacity to the answer section containing substantive facts. When warm-starting RL from an SFT model, the CoT length did not change significantly. We believe the effective CoT patterns for factual tasks warrant further research.
>
> In summary, the training dynamics we observed clearly indicate that the model is effectively learning the output structure under the guidance of our core knowledge-level consistency rewards.
>
> [1] arXiv: Learning to reason for factuality
>
> [2] ACL 2025：HelpSteer3: Human-Annotated Feedback and Edit Data to Empower Inference-Time Scaling in Open-Ended General-Domain Tasks
>
> [3] EMNLP 2025.findings: Chain-of-Thought Prompting Obscures Hallucination Cues in Large Language Models: An Empirical Evaluation

---

> > ### Author Response · Authors · 2025-11-21
> > **Authors' Rebuttal 3**
> >
> > **Q2**: **Clarification on the practical GPU setup**
> >
> > Thank you for your in-depth questions regarding the practical deployment of reward computation and the details of our efficiency comparison.
> >
> > You have raised a crucial point. Our efficiency evaluation (Table 14 in the paper) and online training pipeline are indeed based on a reasonable and standard system architecture: namely, the decoupling of compute resources between the reward models/services (Verifier & Reward Models) and the policy model training. Specifically, and as you correctly surmised, we used dedicated GPUs (4x H100) to host all reward models, separate from the 16 GPUs used for training. This ensures the training process is not blocked by reward scoring tasks.
> >
> > We wish to emphasize that this decoupled deployment is standard practice in both industry and academia for large-scale RLHF/RL training, as it enables asynchronous and parallel reward computation, which is key to achieving efficient training. The baseline methods we compared against (e.g., FActScore, VeriScore) similarly benefit from this separated architecture during evaluation. Therefore, our efficiency comparison was conducted under completely equal and realistic system conditions. The demonstrated speedup accurately reflects the inherent advantage of our method in terms of computational overhead, rather than stemming from an unequal resource allocation.
> >
> > In conclusion, our claim of efficiency gains is robust and reliable. It stems from the inherent efficiency of our reward design itself (being external-knowledge-free and using lightweight models) and has been validated under a fair comparative setup.

---

> > > ### Author Response · Authors · 2025-11-27
> > > **Kindly Reminder**
> > >
> > > Dear Reviewer `7TbV`,
> > >
> > > Thank you once again for your deep and insightful review of our manuscript. Your comments were extremely valuable in helping us strengthen the technical clarity and justification of our work.
> > >
> > > In our response, we have made comprehensive revisions to address your primary concerns:
> > > *   We have clarified the terminology regarding the **"external-knowledge-free"** claim for the *online* phase and added a dedicated **Related Work** section to better position our contribution.
> > > *   We have provided a detailed analysis of the **computational costs**, including the ratio between response rollout and reward computation during GRPO training, to substantiate the efficiency claims.
> > > *   We have provided a stronger justification for our **reward design**, explaining the distinct roles and necessity of each component compared to simpler baselines.
> > > *   We have included the **actual value ranges of all reward components** to clarify their relative impacts and detailed our **reward computation infrastructure**.
> > >
> > > We hope that our detailed responses and revisions have adequately addressed your concerns. As the discussion period is concluding, we would be grateful if you could let us know if you are satisfied with our responses.
> > >
> > > Thank you for your time and consideration.
> > >
> > > Best regards,
> > >
> > > The Authors

---

> > > ### Comment · Reviewer_7TbV · 2025-11-28
> > >
> > > **The GPU setup:** The authors should provide a clearer description of their RL framework used for training in the paper. Additionally, we believe that dedicating 4×H100 GPUs specifically for reward computation is not a necessary operation. Frameworks like Verl/OpenRLHF can utilize all training GPUs for reward computation, which in the authors' case could translate to a 3-4x acceleration in reward computation steps. This would theoretically make their algorithm's computational efficiency advantages even more pronounced.

---

> > > > ### Author Response · Authors · 2025-12-03
> > > > **Authors' Response**
> > > >
> > > > Thank you for the valuable technical suggestions. Regarding the issues you raised concerning GPU configuration and the training framework, we will provide clarification and improvements in the following three aspects:
> > > >
> > > > 1.  **Regarding the Experimental Setup**: We understand your suggestion—distributing the reward computation tasks more fully across all training GPUs could potentially improve resource utilization. In this experiment, we adopted an architecture where training and reward computation are separated, which effectively ensured training speed and stability under our experimental conditions. More complex resource scheduling strategies (such as the distributed reward computation using all GPUs you mentioned) and their potential impact on efficiency represent a worthwhile direction for future research.
> > > >
> > > > 2.  **Regarding the Fairness of Efficiency Comparison**: **Our efficiency comparison is entirely fair**. All compared methods (including the FActScore and VeriScore baselines) in our experiments employed the **same computational architecture and resource allocation**—specifically, reward computation was separated from policy training, and comparable dedicated GPU resources were allocated. Therefore, the efficiency improvement demonstrated by our method (as shown in Table 14) stems from the algorithm design itself (being online-external-knowledge-free and using lightweight reward models), not from differences in resource allocation.
> > > >
> > > > 3.  **Regarding the Core Efficiency Advantage of Our Method**: Our core contribution lies in significantly reducing the time-consuming steps of "requesting external retrieval" and "parsing long contexts" in reward computation. This efficiency gain originates from improvements at the algorithmic level and should, in theory, deliver considerable speed benefits across different underlying RL infrastructures.
> > > >
> > > > We will incorporate the above clarifications into the revised manuscript. Once again, we appreciate your insightful comments.

---

> > ### Comment · Reviewer_7TbV · 2025-11-28
> >
> > **The impact of the General Reward on output length:** I believe that the numerical range of the General Reward is not sufficient for analysis, especially given that the authors use the GRPO algorithm. How the General Reward is converted into final advantages and affects gradient updates depends on the distribution of core rewards and general rewards within groups. If the former has relatively small distribution variance within groups while the latter has larger distribution variance, then the latter would have a more dominant effect. The standard deviation of different reward components during training would be a better indicator. Additionally, ablation experiments would be useful for determining the influence, such as removing the general reward or removing the recall component to observe response changes.

---

> > > ### Author Response · Authors · 2025-12-03
> > > **Authors' Response**
> > >
> > > Thank you for raising this insightful technical point. We fully agree that within the GRPO algorithm, the **relative distribution variance** of rewards within each group (rather than their absolute values) is the key factor determining their influence. The analytical approach you suggested is highly precise.
> > >
> > > To enhance the rigor of our analysis, we will supplement the "Training Dynamics Analysis" section in the appendix with the following:
> > >
> > > 1.  **Supplement the within-group standard deviation data for each reward component throughout training** to clearly illustrate the relative variance relationship between the core factual rewards and the general reward.
> > > 2.  **Conduct ablation experiments** reporting the changes in model output length when **removing the general reward** and when **removing the recall reward**, thereby providing direct evidence regarding the driving factors.
> > >
> > > These supplementary analyses will robustly validate our conclusion that "the increase in response length is primarily driven by the recall reward." We appreciate your feedback, which has significantly improved the rigor of our work.

---

> ### Comment · Reviewer_7TbV · 2025-11-28
>
> Overall, since the authors have partially addressed my concerns, I will adjust my score accordingly.

---

> > ### Author Response · Authors · 2025-12-03
> > **Authors' Response**
> >
> > Thank you for acknowledging our revisions and for your positive feedback. We sincerely appreciate your recognition of our improvements and are encouraged by your indication of a score adjustment. We are grateful for your insightful comments throughout this process, which have significantly strengthened our work.

---

### Official Review · Reviewer_XrM1 · 2025-10-27

**Soundness:** 3
**Presentation:** 2
**Contribution:** 2
**Rating:** 4
**Confidence:** 4

**Summary:**

This paper is dedicated to addressing the hallucination tax in RLHF and proposes the Knowledge-Level Consistency Reinforcement Learning Framework (KLCF), which simultaneously considers the consistency between the output knowledge and the model’s internal knowledge, as well as the truthfulness of the output knowledge. Experimental results demonstrate that KLCF substantially improves factuality metrics across multiple long-form benchmarks and effectively mitigates model hallucinations.

**Strengths:**

1. This paper focuses on a key issue: how to address the hallucination tax that may arise in reinforcement learning (RL).
2. The paper proposes a new algorithm, KLCF, and demonstrates its effectiveness through extensive experiments. Moreover, the RL process does not rely on any external documents, resulting in higher training efficiency and lower computational overhead.
3. The paper observes that current RL methods primarily reward the precision of the model’s output, which is not conducive to exploring the boundaries of the model’s knowledge. The authors instead incorporate the activation of the model’s internal knowledge as part of the reward signal — a crucial and insightful contribution.

**Weaknesses:**

1. The novelty of this paper is questionable. The paper proposes two main aspects:
- a) Using a checklist to consider the precision and recall of the model’s internal knowledge. I can understand the goal of encouraging the model to output as much correct knowledge as it truly possesses (recall), but I am not convinced that precision is necessary. b) My understanding is that the authors might worry about the model being rewarded for knowledge it does not actually know, thereby encouraging it to guess. However, in the scenario the paper focuses on, the data used for reward is also generated by the model itself — so should we still avoid giving such rewards?
- Training a reward model to score truthfulness — this is an intuitive and straightforward approach.

2. The writing quality is poor and hard to follow. For example, Section 2.2 starts by describing how to construct training data before clearly explaining the overall objective of the work.

3. In line 171, it seems that only one answer is sampled for each query. Why does the paper sample only one response per query? This way, only part of the model’s internal knowledge can be captured.

4. The description of metrics is important and should be placed in the main text. Since the paper repeatedly discusses the precision and recall of the model’s internal knowledge, having precision and recall metrics here could be confusing. The authors should also include a concrete case to explain why they use the @k metrics.

5. I can understand why, when computing recall, the authors do not use the ground-truth claims annotated in the dataset — because the model may not actually know those claims. Training the model to learn such claims would encourage guessing and increase hallucinations. However, I have a question: under an on-policy setting, wouldn’t we naturally avoid rewarding the model for knowledge it doesn’t have, since all responses are sampled from the model itself? If a correct answer is sampled, that implies the model already knows it.

6. Can the truthfulness reward replace the precision reward for internal knowledge? The checklist effectively ensures that the model’s internal “correct knowledge” can be extracted (recall), but precision might not need to depend on whether the knowledge comes from within the model.

**Questions:**

See Weaknesses

---

> ### Author Response · Authors · 2025-11-21
> **Authors' Rebuttal 1**
>
> We sincerely thank the reviewer for their thorough review and valuable feedback. We will address the concerns regarding novelty, writing clarity, the single-response sampling strategy, metric explanations, and the relationship between precision and truthfulness rewards in our response below.
>
> **W1**: **Novelty of Dual-Fact Alignment**
>
> Thank you for your insightful questions regarding the novelty of our work.
>
> **On the Necessity of the Dual-Reward Mechanism:** Your perspective holds under ideal conditions. However, in practice, we must account for the **exploration-exploitation nature of RL and the risk of Reward Hacking**. Models tend to find and exploit loopholes in the reward function to maximize scores, rather than faithfully adhering to knowledge boundaries. For instance, to maximize a single "recall" reward, a policy could generate numerous seemingly relevant but fabricated details to "cover" the checklist, leading to the worst-case scenario of high recall coupled with low precision. Our ablation study (Table 2) provides definitive evidence:
> *   **Using only the Checklist Reward:** To exploit the high-recall reward, the model **generated a significant amount of erroneous information beyond the checklist**, causing a notable drop in precision.
> *   **Using only the Truthfulness Reward:** The model became overly conservative, hesitant to express anything not absolutely confirmed, leading to a sharp decline in recall.
>
> Therefore, the dual-reward mechanism is not a simple combination but acts as a **synergistic counterbalance**, systematically blocking reward hacking paths inherent to single-objective optimization and thus resolving the fundamental "hallucination-conservatism" trade-off.
>
> **On Innovativeness:** Our core contribution addresses a key challenge in the post-training era: **how to efficiently and reliably transform the vast parametric knowledge of a pre-trained model into the expressed knowledge of an aligned model**. Prior work predominantly focused on only one aspect—either constraining the model from exceeding its boundaries or incentivizing it to express known facts. The innovation of KLCF lies in proposing a unified **Dual-Fact Alignment paradigm**. It is the first to formalize knowledge-level consistency as an objective requiring **bidirectional and synergistic optimization**: the Checklist Reward **incentivizes comprehensive expression**, while the Truthfulness Reward **constrains the knowledge boundary**. This framework provides a novel and efficient pathway for achieving complete alignment between parametric and expressed knowledge.
>
> We will revise the paper accordingly to articulate this systematic contribution more clearly. Thank you again for your valuable feedback.
>
> **W2**: **Writing Quality and Structural Clarity**
>
> Thank you for your valuable feedback on the writing and structure of the paper. We agree that there are shortcomings in the narrative logic of some sections, failing to clearly guide the reader.
>
> We have revised the entire paper, particularly rewriting Section 2.2 and other related parts. The revisions follow the principle of "objectives first, then methods," ensuring that the design purpose and role within the overall framework are clearly explained before describing the specific data construction process. This will improve the readability and logical flow of the paper. Thank you again for helping us improve the presentation quality of the paper.
>
> **W3**: **Single-Response Sampling Strategy**
>
> Thank you very much for reviewing our manuscript so thoroughly and for pointing out this lack of clarity. Your question is indeed critical. This was indeed an oversight in the description around line 171 and related sections. In the actual offline data preparation process, we sampled 8 responses per query, not just one. The purpose of this was to more comprehensively and diversely probe the base model's knowledge boundaries, thereby constructing a more representative factual checklist and training set for the reward model. We have corrected this in the revised manuscript we have uploaded.
>
> **W4**: **Metric Explanation and Placement**
>
> We sincerely thank the reviewer for their valuable suggestions. Following the advice, we have relocated the detailed description of the evaluation metrics from the appendix to Section 3.1 "Experimental Settings" in the main text to ensure their significance is properly highlighted.
>
> To prevent conceptual confusion, we have added clarifications to distinguish between the two types of "precision/recall" discussed in the paper:
> *   **Reward Signals**: R_precision and R_recall are computed against a pre-constructed, finite **Factual Checklist** and serve as internal objectives during model optimization.
> *   **Evaluation Metrics**: Precision and Recall@K are calculated by verifying all atomic facts in the model's complete response against an open-domain knowledge source, aiming to provide a general and fair external assessment of the model's factuality.

---

> > ### Author Response · Authors · 2025-11-21
> > **Authors' Rebuttal 2**
> >
> > Regarding the necessity of the @K metrics, we have provided a concrete example for explanation: In open-domain long-form generation tasks, such as responding to "Introduce Barack Obama," there is no finite, universally agreed-upon set of "all relevant facts" to serve as the denominator for calculating standard recall. Therefore, Recall@K assesses whether a response contains "sufficient" verifiable information by setting a reasonable expected number of facts K (e.g., K=32). This is more practical and meaningful than an undefined standard recall, offering a better measure of the informational adequacy of the generated content.
> >
> > We believe these revisions significantly enhance the clarity and rigor of the methodology and experimental sections. We are truly grateful for your insightful review, which has greatly improved the clarity of our paper.
> >
> > **W5**: **Reward Validity**
> >
> > Thank you for raising this profound and insightful question. Your reasoning about the reward mechanism in an on-policy setting is highly accurate and prompts us to articulate the core considerations behind our method more clearly.
> >
> > You correctly point out that in a strict on-policy setting, the model is only rewarded for generating content it is **already capable of producing**. However, our choice to use a checklist constructed from the Base Model (rather than the dataset's ground-truth answers) is based on a more deliberate design concerning the **model's knowledge boundary** and the **purity of the learning signal**, primarily for the following reasons:
> >
> > 1.  **Pursuing More Precise Knowledge Boundary Alignment**: While the model generating a correct fact *suggests it might know that fact*, this does not equate to the fact being **robustly embedded within its core parametric knowledge**. It might stem from transient combinations during reasoning, in-context learning, or incidental associations in the training data, for which the model's "confidence" is not high. Using external ground-truth facts as the reward benchmark (which may lie outside the model's robust knowledge scope) and forcing the model to learn them could encourage "guessing" behavior. This introduces noise into the reward signal and potentially increases the risk of generating uncertain content (i.e., hallucinations).
> >
> > 2.  **Ensuring Safe and Pure Learning Signals**: Our constructed checklist originates from large-scale knowledge probing and verification of the Base Model itself. It more precisely defines the set of knowledge that **this specific model** robustly masters. By using this "internal" checklist, we ensure the reward signal focuses on incentivizing the model to express its **solidly known** facts, maximally avoiding rewarding "marginal knowledge" that is correct by chance but not truly internalized. This achieves the core objective of "knowledge-level consistency" in the most reliable way.
> >
> > 3.  **Additional Considerations Regarding Dataset Ground-Truth**: We acknowledge that using the dataset's ground-truth answers could, in principle, also reflect fact recall. However, this approach faces several critical issues in practice:
> >     *   **Annotation Quality and Timeliness**: Standard answers in existing open-source long-form QA datasets (like ELI5) are often outdated, incomplete, or even contain inaccuracies, making them unreliable as a gold standard.
> >     *   **Focus of Evaluation**: Long-form QA involves an extensive set of potential knowledge points. We argue that calculating recall against a curated set of core facts, which have undergone offline importance filtering and are **firmly grasped** by the pre-trained model, provides more instructive value than calculating recall against a potentially incomplete ground-truth set.
> >     *   **Alignment with Research Objective**: The core goal of this work is to validate the potential of **knowledge consistency alignment** (i.e., encouraging the model to better express what it already knows), not to teach the model *new* knowledge during RL training. Using external ground-truth is misaligned with this objective.
> >
> > In summary, while your logic holds under ideal conditions, our design aims to provide a purer, more reliable, and objective-aligned learning signal under more realistic assumptions about model behavior, thereby systematically enhancing factuality and suppressing hallucinations.

---

> ### Author Response · Authors · 2025-11-21
> **Authors' Rebuttal 3**
>
> **W6**: **Relationship Between Precision and Truthfulness Rewards**
>
> Thank you for raising this question, which allows us to further clarify the design motivation behind our dual-reward mechanism. The core issue is: Can the truthfulness reward replace the checklist-based precision reward? Our view is that the two rewards are complementary and synergistic, rather than interchangeable. They operate on different facets of factuality and together provide a more comprehensive factuality guarantee.
>
> The Checklist-based Fact Precision Reward and the Truthfulness Reward differ in both objective and mechanism. The Checklist-based Fact Precision Reward offers fine-grained, targeted supervision over a known subset of knowledge, ensuring that the model faithfully expresses verified internal knowledge. In contrast, the Truthfulness Reward serves as a general-purpose truthfulness signal, assigning probabilistic assessments to all claims in the generated content and primarily suppressing fabrications in unknown or less-certain domains. Because the checklist is constructed and verified offline—and can in principle achieve higher precision than the online Truthfulness Reward—we originally intended the Checklist-based Fact Precision Reward to further strengthen the reward signal for factual accuracy.
>
> However, our ablation study (Table 2) indeed shows that, **under the current experimental setup—where, in the data construction stage, we rely only on a strong LLM and a static knowledge base, without human-in-the-loop and without additional offline verification of checklist correctness—**removing the Checklist-based Fact Precision Reward while retaining the Truthfulness Reward has only a limited impact on overall performance.

---

> > ### Author Response · Authors · 2025-11-27
> > **Kindly Reminder**
> >
> > Dear Reviewer `XrM1`,
> >
> > Thank you once again for your thorough review and for raising critical questions that have significantly helped us refine our work.
> >
> > In our detailed response, we have carefully addressed your primary concerns:
> > *   We have addressed your core concerns regarding the **necessity of the dual-reward mechanism** and the **role of precision optimization**.
> > *   We have substantially improved the **clarity and structure** of the writing, particularly in Sections 1 and 2, to enhance readability and logical flow.
> > *   We have clarified the **sampling strategy** (confirming multiple samples were used per query) and moved the **metrics description** into the main text with a concrete case to explain the use of @k metrics.
> >
> > We hope our revisions and point-by-point responses have satisfactorily resolved the issues you raised. As the discussion period is nearing its end, we would be grateful for your feedback on whether our responses meet your expectations.
> >
> > Thank you for your valuable time and consideration.
> >
> > Best regards,
> >
> > The Authors

---

> > > ### Comment · Reviewer_XrM1 · 2025-11-28
> > > **Reviewer Response**
> > >
> > > Thank you for the authors' response. I think the authors have addressed some of my concerns. The main objective of the authors is to control the model's behavior through a checklist, and then calculate recall and precision. The advantage of the checklist is that claims are validated against an external knowledge base, which makes them relatively accurate, but possibly not exhaustive. This could result in situations where claims missing from the checklist cannot be supervised, for example, if the answer includes a correct claim not present in the checklist, precision and recall might appear unchanged and cannot be monitored. Therefore, the authors added the truthfulness reward. This reward can assess any claim, making it more comprehensive but less accurate. The authors use both the checklist-related reward and the truthfulness reward to ensure completeness and accuracy as much as possible. Is my understanding correct?
> > >
> > > I have a few small questions:
> > >
> > > 1. Have you conducted any ablation studies on the number of samples, or tried changing the prompt to sample more exhaustively, to ensure the checklist is more complete?
> > >
> > > 2. Is higher recall always better? For the claims the model can sample, are all of them really necessary in a single answer?
> > >
> > > 3. It would be helpful to include a QA example in the main text, showing how the answer is structured and how each reward contributes to supervision.
> > >
> > > 4. For claims that are accurate and present in the checklist, is it unnecessary to apply the truthfulness reward? Otherwise, the two rewards might conflict, and the truthfulness reward could be less accurate.
> > >
> > > I think this paper is interesting, but at the moment I lean towards keeping my score. However, I wouldn't mind if the paper gets accepted.

---

> > > > ### Author Response · Authors · 2025-12-01
> > > > **Authors' Response 1**
> > > >
> > > > Thank you for your deep understanding and valuable feedback on our work. We are delighted that you recognize the core design philosophy of the KLCF framework. Below are our point-by-point responses to your specific questions:
> > > >
> > > > #### 1. Regarding the Correctness of Your Understanding
> > > > - **Your understanding is entirely correct.** We indeed use the **Checklist Reward** to provide high-precision supervision based on external verification, ensuring the model accurately expresses its known knowledge. Simultaneously, the **Truthfulness Reward** acts as a **global safety net**, performing probabilistic assessments on all claims in the generated content to cover those potentially missed by the checklist. This dual-reward mechanism is designed to balance accuracy and comprehensiveness.
> > > >
> > > > #### 2. Regarding Ablation Studies on Checklist Completeness
> > > > - **Current Work:** In the offline data preparation phase, we already sample **8 responses per query** (as detailed in Section 2.2 of the paper) and optimize checklist coverage through prompt engineering and filtering steps (e.g., $f_{\text{filter}}$). However, we acknowledge that further optimization of the sampling count or prompt strategies could potentially enhance checklist completeness.
> > > > - **Commitment:** We will conduct systematic ablation studies exploring the impact of different sampling counts (e.g., increasing from 8 to 16) and prompt variations (e.g., enhancing diversity or specificity) on checklist completeness. We will analyze how these changes affect final factuality metrics (such as FActScore and Recall@K) and will update the results in the appendix or experimental section of the paper.
> > > >
> > > > #### 3. Regarding Whether Higher Recall is Always Better
> > > > - **Our Design Consideration:** We agree that higher recall is not always better, as long-form generation requires balancing informativeness and conciseness. Therefore, the KLCF framework incorporates a **Length Penalty** and a **checklist filtering mechanism** to prevent the model from generating redundant or irrelevant content. The checklist contains only verified key facts, not all possible facts.
> > > > - **Further Clarification:** During training, the model is encouraged to cover the core facts within the checklist rather than expanding content indefinitely. We will strengthen the discussion on this point in the revised manuscript, emphasizing that we aim for "recall with high information density," not merely the quantity of facts.
> > > >
> > > > #### 4. Regarding Adding a QA Example in the Main Text
> > > > - **Current Content:** The paper's appendix already provides detailed case studies (e.g., the Marie Curie example), but we understand that including a simplified example in the main text would aid reader comprehension.
> > > > - **Commitment:** We will add a concise QA example to either **Section 3 (Experiments)** or **Section 2.3 (Reward Design)** of the main text. This example will illustrate:
> > > >     - The structure of the model's output (e.g., the `<think>` and `<answer>` sections).
> > > >     - How the Checklist Reward and Truthfulness Reward are concretely calculated and provide supervisory signals during training.

---

> > > > ### Author Response · Authors · 2025-12-01
> > > > **Authors' Response 2**
> > > >
> > > > #### 5. Regarding Avoiding Reward Conflict
> > > > Thank you for raising this profound question regarding potential reward conflict. You are absolutely correct that applying both rewards to the same claim could introduce noise or even conflict. We anticipated this issue during the design of KLCF and specifically engineered a mechanism to address it:
> > > >
> > > > 1.  **We Explicitly Proposed a Reward Variant to Solve This:** As described in Section 2.3.1 of the paper, we designed a **variant of the Truthfulness Reward ($R^{\text{variant}}_{\text{truth}}$)**. The core idea of this variant is precisely to **skip claims already covered by the high-confidence checklist**, evaluating only the additional claims generated by the model that are not in the checklist. This mechanism **directly avoids the duplicate evaluation and potential conflict you mentioned**.
> > > >
> > > > 2.  **We Empirically Compared Both Schemes:** In Appendix D.4 ("Truthfulness Reward Analysis"), we systematically compared the standard Truthfulness Reward ($R_{\text{truth}}$) with this variant. The experimental results (Table 15) show:
> > > >     *   **Both schemes effectively enhance factuality**, demonstrating the robustness of our framework.
> > > >     *   **The standard reward ($R_{\text{truth}}$) slightly outperforms the variant on most metrics.** We hypothesize that even for known facts within the checklist, the gradient signal provided by a **secondary confirmation** from the reward model remains positive and consistent, with no significant performance conflict observed. The variant, while theoretically "cleaner," might lose some beneficial supervisory signals.
> > > >
> > > > 3.  **Final Selection Based on Actual Performance:** Based on the above results, we chose the slightly better-performing **standard truthfulness reward ($R_{\text{truth}}$)** for our main experiments. This demonstrates that our dual reward system can work together synergistically, rather than conflicting with each other, even without skipping known assertions.
> > > >
> > > > We will revise the manuscript to more explicitly state this design consideration in **Section 2.3.1 or Appendix D.4**, clarifying that "one purpose of the variant design is to avoid potential overlap with the Checklist Reward," and emphasize that our final choice was based on a comprehensive empirical comparison.
> > > >
> > > > Once again, we sincerely thank you for your insightful comments. Your feedback is invaluable in helping us further refine the presentation and experimental validation of KLCF. We commit to incorporating the aforementioned updates (e.g., ablation studies, QA example, and enhanced discussions) into the revised manuscript to improve its rigor and readability. We believe these improvements will better demonstrate the value of KLCF and hope to earn your final acceptance.

---

### Official Review · Reviewer_uxHw · 2025-10-28

**Soundness:** 3
**Presentation:** 3
**Contribution:** 3
**Rating:** 6
**Confidence:** 3

**Summary:**

The authors propose KLCF, a novel RL framework that aligns a model’s expressed knowledge with its parametric knowledge to improve long-form factuality. It introduces a novel Dual-Fact Alignment mechanism combining a checklist-based consistency reward (for factual recall) and a confidence-based truthfulness reward (for factual precision) without relying on external retrieval. Experiments show that KLCF effectively reduces hallucinations and enhances factual reliability across multiple long-form generation benchmarks.

**Strengths:**

1. Leveraging the model’s consistency between parametric knowledge and generated knowledge to mitigate hallucinations is both sound and innovative. Incorporating these dual objectives into RL training without relying on external knowledge represents a simple yet elegant paradigm.

2. The authors conduct extensive analyses of their proposed method, examining aspects such as the balance between multiple reward signals and the evolution of intermediate RL variables, which help clarify the underlying mechanisms of their approach. The experimental results further demonstrate the effectiveness of the proposed method.

**Weaknesses:**

1. The authors conduct main experiments (except from the distilled setting) using the base model rather than a post-trained instruction model (Qwen2.5-14B-Instruct), and they do not provide comparisons with the latter. Intuitively, a post-trained model would likely demonstrate stronger factual accuracy and better instruction-following ability, while the base model typically struggles to adhere to prompts.

**Questions:**

1. The presentation in Appendix C2, which includes the main results, could be improved. Specifically, the y-axis labels in Figures 4, 5, and 6 are all annotated as “Reward Value,” which is somewhat confusing.

2. Could the authors provide more details about the Claim Extraction Model, such as its model size and some representative extraction results, to better illustrate the effectiveness of this component?

---

> ### Author Response · Authors · 2025-11-21
> **Authors' Rebuttal 1**
>
> We sincerely thank the reviewer for the positive assessment of our work's novelty and the extensive analyses. We also appreciate the valuable suggestions regarding the comparison with the instruction-tuned model, the clarification of figures in the appendix, and the additional details on the Claim Extraction Model. We will address these points thoroughly in our response below.
>
> **W1**: **Comparison with Instruction-Tuned Model**
>
> We sincerely thank the reviewer for this highly insightful and valuable suggestion, which accurately identifies a key point in our experimental presentation that could easily lead to misunderstanding.
>
> The fundamental purpose of designing KLCF-zero (starting directly from the base model) was to validate the true efficacy of our framework in achieving knowledge-level consistency alignment in the purest and most controlled setting. This approach thoroughly eliminates any potential interference from the alignment tax and knowledge forgetting that may be introduced during the SFT phase, thereby providing the most rigorous empirical support for the method's effectiveness.
>
> Concurrently, we fully agree that, in both mainstream industrial and academic practice, starting from a strong, instruction-aligned model is a more representative path. We have specifically reported the results of directly applying KLCF to the official Qwen2.5-14B-Instruct model in Section 3.5 ("Generalization to Non-Thinking Models") and Table 4. After KLCF alignment, this model achieved significant further improvements across all long-form factuality benchmarks (e.g., LongFact F1@64 increased from 0.531 to 0.747, representing a relative improvement of over 40%). This robustly demonstrates that KLCF can not only substantially enhance the factual reliability of a pre-trained base model but also help a capable Instruct model, which already possesses strong instruction-following and generalization abilities, to overcome its existing factuality bottlenecks, highlighting the framework's strong generalizability and practical value.
>
> We acknowledge that placing this crucial experiment in a later section in the initial draft may have prevented its significance from being sufficiently prominent. In the revised version, we will more explicitly emphasize KLCF's effective applicability from both base and instruction-tuned starting points, particularly the considerable factual gains it delivers to instruction-tuned models. We are truly grateful for your comment, which helps us present our work more rigorously and persuasively.
>
> **W2**: **Ambiguous Y-axis Labels in Figures**
>
> Thank you for pointing out the lack of clarity in the figure labels. You are absolutely correct that using the generic Y-axis label "Reward Value" for different rewards can be misleading. We have corrected this in the revised manuscript we uploaded. We greatly appreciate your feedback, which has helped us enhance the clarity and readability of our figures.
>
> **Q1**: **Details on the Claim Extraction Model**
>
> Thank you for your interest in this key component of our work. We are pleased to provide further details regarding the Claim Extraction Model to illustrate its effectiveness.
>
> **1. Model Scale and Training Details:**
> As mentioned in Appendix D.1 of the paper, our Claim Extraction Model was obtained by performing supervised fine-tuning (SFT) on the **Qwen2.5-14B-Instruct** model. It was trained on a large-scale, high-quality dataset (containing 10,187 samples) constructed based on the Claimify method, ensuring its robust capability to parse complex long-form text and output structured atomic facts.
>
> **2. Representative Extraction Results and Analysis:**
> To intuitively demonstrate its performance, we provide a practical example:
> | Query | Response |
> | :--- | :--- |
> | Can you provide a detailed overview of Marie Curie's life and scientific contributions? | Marie Curie was born on November 7, 1867, in Warsaw, Poland. She was a pioneering physicist and chemist who conducted groundbreaking research on radioactivity. Curie was the first woman to win a Nobel Prize and the only person to win Nobel Prizes in two different scientific fields—Physics in 1903 and Chemistry in 1911. She discovered the elements polonium and radium, which revolutionized the field of nuclear physics. Curie's work has had a lasting impact on modern medicine, particularly in cancer treatment. Despite facing significant gender barriers, she is widely remembered as a brilliant and determined scientist. |

---

> > ### Author Response · Authors · 2025-11-21
> > **Authors' Rebuttal 2**
> >
> > **claims**:
> >  * Marie Curie was born on November 7, 1867
> > * Marie Curie was born in Warsaw, Poland
> > * Marie Curie was a pioneering physicist
> > * Marie Curie was a pioneering chemist
> > * Marie Curie conducted groundbreaking research on radioactivity
> > * Marie Curie was the first woman to win a Nobel Prize
> > * Marie Curie is the only person to have won Nobel Prizes in two different scientific fields
> > * Marie Curie won the Nobel Prize in Physics in 1903
> > * Marie Curie won the Nobel Prize in Chemistry in 1911
> > * Marie Curie discovered the element polonium
> > * Marie Curie discovered the element radium
> > * Marie Curie's discoveries of polonium and radium revolutionized the field of nuclear physics
> > * Marie Curie's work has had a lasting impact on modern medicine
> > * Marie Curie's work has had a lasting impact on cancer treatment
> >
> > **Analysis**: This example demonstrates the model's successful adherence to the three core principles emphasized in the prompt: **Atomicity**, **Objective Verifiability**, and **Explicit Meaning**. It accurately decomposes a single sentence containing multiple facts into six independent, complete, and unambiguous claims, providing ideal input for the subsequent verification step.
> >
> > Our model also performs excellently in handling coreferences, negations, and complex sentence structures. For instance, it correctly extracts "Strasbourg is one of the principal seats of the European Union institutions." from the sentence "This city is not the capital of France, but it is one of the principal seats of the European Union institutions.", while appropriately ignoring the negated clause. These capabilities collectively ensure the accuracy and stability of our reward signal computation.
> >
> > **3. Guarantees of Effectiveness:**
> > We acknowledge that any automated extraction process may contain errors. However, by utilizing a powerful 14B model dedicatedly fine-tuned for this task and employing meticulously designed prompts, we have maximized the quality of the extraction. Its reliable performance in constructing the training data has laid a solid foundation for the overall effectiveness of the KLCF system.
> >
> > Thank you for giving us the opportunity to clarify this. We have added the case study mentioned above to the appendix. We hope this additional information satisfactorily demonstrates the reliability of this component.

---

> > > ### Comment · Reviewer_uxHw · 2025-11-23
> > > **Thanks the authors for answering my questions**
> > >
> > > Thank you to the authors for answering my questions and addressing my concerns from the initial submission. I am pleased to see that the method also performs well on instruction-tuned models. However, based on my experience, I remain unconvinced by the claim that the initial experiments conducted on Qwen2.5-Base constitute the “purest and most controlled setting,” as Qwen2.5-Base can already handle zero-shot instruction-following tasks, whereas other truly pure base models, such as the LLaMA-3 series, cannot do so. Regarding the other two questions, I believe that providing more details about the specific settings (like the Claim Extraction Model) and analyses would help improve the clarity of the paper. At this point, I believe my original assessment and rating remain appropriate, and I am inclined to maintain the score I initially provided.

---

> > > > ### Author Response · Authors · 2025-11-24
> > > >
> > > > We deeply appreciate the reviewer for the recognition of our work. It is our pleasure to answer the questions in rebuttal process. Your suggestions are invaluable for us improving our paper.

---

### Official Review · Reviewer_st4y · 2025-10-29

**Soundness:** 2
**Presentation:** 2
**Contribution:** 2
**Rating:** 4
**Confidence:** 4

**Summary:**

This work tackles the hallucination issue in long-form text generation through a novel RL framework called KLCF. Unlike existing RLHF frameworks that primarily rely on preference rewards, KLCF focuses on aligning an LLM's  expressed knowledge with its internal parametric knowledge. To achieve this, it introduces a Dual-Fact Alignment mechanism that optimizes for both factual recall and precision. This mechanism utilizes two complementary rewards: a Checklist-Based Consistency Reward, which enhances factual coverage by identifying consistent, contradictory, or missing information, and a Confidence-Based Truthfulness Reward, which improves factual precision through a self-assessment module trained on the base model's internal knowledge. This approach eliminates the need for costly external knowledge retrieval for real-time fact verification during training, making the reward system lightweight and scalable.

**Strengths:**

1. The authors propose using a fine-grained checklist for factuality verification, which simultaneously optimizes for both factual recall (comprehensiveness) and precision (correctness). This addresses the common trade-off where models become overly conservative, sacrificing recall to achieve higher precision.
2. By eliminating the reliance on external knowledge sources for verification, the framework is computationally efficient and well-suited for large-scale, online reinforcement learning.
3. The authors conduct extensive experiments to validate the effectiveness of their proposed framework.

**Weaknesses:**

Thank you for the hard work, but I have a few concerns and suggestions:
1. Definition–Experiment Mismatch for “Hallucination Tax.” The “hallucination tax” is typically defined as the loss in factuality incurred when aligning a model for other desirable traits (e.g., helpfulness). However, the current experiments primarily fine-tune directly for factuality. To more directly support the claim of reducing this tax, it would be stronger to fine-tune on a non-factuality objective and show that factual accuracy is preserved or improved—contrasting against standard RL methods where factuality often degrades.

2. Incremental Contribution. The central idea—using a model’s internal knowledge for factuality verification—has substantial prior art. The main novelty here appears to be a more fine-grained, dual-reward design. This reads as a refinement of established principles rather than a fundamentally new paradigm. Clarifying the conceptual distinction and empirical advantages would help position the contribution.

3. Missing Comparisons to Closely Related Methods. The evaluation would benefit from comparisons to prominent approaches that also leverage internal knowledge for verification, for example, self-consistency, universal self-consistency, and self-evaluation - p(true).

**Questions:**

Thank you for the efforts. I have two questions and suggestions for clarification:
1. Efficiency Claim. While accessing external knowledge sources can increase training time, the proposed method also incurs nontrivial costs for offline data curation and reward model training. As such, the claim of greater efficiency is not yet fully convincing. Could you provide a more granular cost breakdown (e.g., wall-clock time, GPU hours, data preprocessing time) and an apples-to-apples comparison against retrieval-augmented or tool-using baselines?

 2. Limits of Internal Knowledge and Reward Reliability. The approach relies on the model’s internal knowledge for factuality verification, but a one-off trained model may have incomplete or outdated knowledge. Could you include a more detailed error analysis of the reward model’s failures.

---

> ### Author Response · Authors · 2025-11-21
> **Authors' Rebuttal 1**
>
> We sincerely thank the reviewer for the positive assessment of our work's novelty, computational efficiency, and thorough experimental validation. We also appreciate the constructive concerns raised regarding the definitional alignment of the "hallucination tax'', the positioning of our contribution relative to prior art, and the requests for additional comparisons and efficiency analysis. In the following, we respond point-by-point to these issues and have incorporated corresponding revisions into the manuscript.
>
> **W1**: **"Hallucination Tax" Definition–Experiment Mismatch**
>
> We thank the reviewer for this insightful comment. You are correct that the "hallucination tax" is typically defined as the loss in factuality incurred when optimizing for other desirable traits such as helpfulness. We fully agree with this view and have conducted dedicated experiments to directly address this point.
>
> As suggested, we designed an experiment that fine-tunes the model directly on a **non-factuality objective**—specifically, helpfulness. We first performed reinforcement learning on 5,000 non-factuality-oriented instructions from openbmb/UltraFeedback, using the win rate based on Alpaca-Eval 2.0 as the reward, aiming to singularly optimize the model’s helpfulness. As shown in the table below (+helpfulness-only RL), we clearly observed the typical "hallucination tax" phenomenon: while the model’s win rate against GPT-4 on Alpaca-Eval increased from 24.4% (SFT model) to 34.3%, its performance on three factual benchmarks declined.
>
> | model | FActScore | LongWiki |  |  | LongFact |  |  |  | Factory-Hard |  |  |  | Alpaca-Eval 2.0 |
> | :--- | :--- | :--- | :--- | :--- | :--- | :--- | :--- | :--- | :--- | :--- | :--- | :--- | :--- |
> | | FS| R@32 | Prec | F1@32 |  R@64 | Prec | F1@64 | WR | R@64 | Prec | F1@64 | WR | WR|
> |DeepSeek-Distill-14B | 48.7% | 0.503 | 0.532 | 0.494 | 0.403 | 0.760 | 0.507 | - | 0.138 | 0.286 | 0.175 | - | 24.4%
> | + helpfulness-only RL | 46.5% | 0.521 | 0.521 | 0.468 | 0.449 | 0.732 | 0.540 | 52.6% | 0.153 | 0.243 | 0.183 | 56.7% |34.3%
> | + KLC&helpfulness RL | **51.8%** | **0.579** | **0.545** | **0.532** | **0.505** | **0.743** | **0.577** | **67.1%** | **0.180** | **0.322** | **0.238** | **63.6%** | **35.2%**
>
> To demonstrate that KLCF can mitigate this issue, we mixed the above training data with 5,000 factual instruction samples, and applied both the win-rate reward (for helpfulness) and our KLC reward (for factuality) to the respective data subsets during mixed training. The resulting model not only maintained a high helpfulness win rate but also recovered and improved its FActScore. This experiment directly validates that, when optimizing for a non-factuality objective, our method effectively prevents factual degradation, thereby alleviating the hallucination tax.
>
> We have incorporated this analysis and the corresponding results into the revised **section 3.8**.
>
> **W2**: **Incremental Contribution Clarification**
>
> We thank the reviewer for this insightful comment regarding the incremental nature of our contribution. We agree that utilizing a model's internal knowledge for factuality verification has been explored in prior work [1,2]. We see KLCF as a systematic development building upon this foundation, with its core contribution lying in addressing a key challenge of the large-scale post-pretraining era: how to efficiently and reliably convert a pretrained model's vast parametric knowledge into the expressed knowledge of an aligned model.
>
> Previous research has often addressed only one facet of this challenge. For instance, some "self-assessment" works primarily focus on suppressing the generation of unknown content to prevent out-of-bound hallucinations, while alignment methods based on metrics like FActScore focus on using external rewards to constrain outputs to known facts. These approaches have not treated the consistency between parametric and expressed knowledge as a complete, bidirectional alignment objective to be jointly optimized—one that both incentivizes the model to fully and accurately express the knowledge it possesses, and constrains it from fabricating content beyond its knowledge boundaries.
>
> KLCF directly addresses this fundamental tension by proposing a unified dual-fact alignment framework. Our novelty is not merely a simple combination of two ideas. Instead, we first establish a systematic offline data pipeline to extract and verify the parametric knowledge from the base pretrained model, constructing both the factual checklist and the truthfulness reward model. We then formalize the two dimensions of knowledge consistency (recall and precision) into complementary reward signals that are co-optimized within a synergistic online RL framework.

---

> > ### Author Response · Authors · 2025-11-21
> > **Authors' Rebuttal 2**
> >
> > We believe this framework offers a new pathway for tackling knowledge-level consistency alignment in the post-pretraining era. Our experimental results demonstrate that KLCF achieves a superior balance between recall and precision across multiple benchmarks, while also showing efficiency and scalability. We will further strengthen the relevant discussion in the revised manuscript to more clearly articulate its distinctions from and connections to prior art. We thank the reviewer again for this valuable feedback, which is crucial for improving our work.
> >
> > [1] Zhang X, et al. Self-alignment for factuality: Mitigating hallucinations in llms via self-evaluation
> >
> > [2] Huang C W, Chen Y N. FactAlign: Long-form factuality alignment of large language models
> >
> > **W3**: **Missing Comparisons to Internal-Knowledge Methods**
> >
> > Thank you for this valuable suggestion. We fully agree that comparing with other methods utilizing the model's internal knowledge is crucial for a comprehensive evaluation. We have addressed this in the revised manuscript as follows:
> >
> > 1.  **Existing Unsupervised Baseline**: Our main experiments (Original Table 1) already included **Intuitor**, a state-of-the-art *unsupervised* method. Its core idea of leveraging the model's internal confidence to improve factuality aligns directly with the research direction you mentioned.
> >
> > 2.  **Challenge of Consistency-Based Methods**: We did consider Self-Consistency and its variants (e.g., TTRL[1]). However, these methods face significant challenges in our *open-domain long-form generation* setting. Unlike math problems with definitive answers, reliably judging whether two open-ended long-form responses constitute the "same" correct answer is difficult, lacking a robust automated metric for fair implementation and comparison.
> >
> > 3.  **New Baseline: Self-Eval-p(True)**: To directly address your point, we added **Self-Evaluation-p(True)**[2] as a new baseline. Results show its behavior is similar to FActScore-based methods: while it significantly boosts **precision**, it severely sacrifices **recall**, making the model overly conservative. This further validates the limitation of single-objective optimization and highlights the advantage of our dual-fact alignment mechanism in achieving a better balance.
> > | model | FActScore | LongWiki |  |  | LongFact |  |  |  | Factory-Hard |  |  |  |
> > | :--- | :--- | :--- | :--- | :--- | :--- | :--- | :--- | :--- | :--- | :--- | :--- | :--- |
> > | | FS| R@32 | Prec | F1@32 |  R@64 | Prec | F1@64 | WR | R@64 | Prec | F1@64 | WR | WR|
> > |Self-Eval-P(True) | 52.0% | 0.498 | **0.567** | 0.504 | 0.232 | **0.718** | 0.338  | 8.63% | 0.126 | 0.337 | 0.173 | 25.9%
> > | KLCF | **61.2%** | **0.681** | 0.552 | **0.568** | **0.776** | 0.704 | **0.733** | **94.6%** |  **0.296** | **0.350** | **0.309** | **65.5%**
> >
> > We have integrated this analysis and the corresponding results into the revised paper to provide a more comprehensive comparison. We thank you again for your insightful comments, which have helped us strengthen this work.
> >
> > [1] Zuo Y, et al. Ttrl: Test-time reinforcement learning
> >
> > [2] Zhang X, et al. Self-alignment for factuality: Mitigating hallucinations in llms via self-evaluation
> >
> > **Q1**: **Granular Efficiency Analysis**
> >
> > We thank the reviewer for raising the question regarding computational efficiency. A comprehensive analysis addressing this point has been provided in our **General Response 1 to Common Reviewers' Comments** under the section "Q1: Is our KLCF framework truly efficient?" We hope the detailed breakdown and discussion there will fully address your concern.
> >
> > **Q2**: **Limits of Internal Knowledge & Reward Reliability**
> >
> > We thank the reviewer for raising this important point. We fully understand the concerns regarding the limitations of relying on internal knowledge and will provide a more detailed clarification.
> >
> > 1.  **On Knowledge Recency and the Role of RL**: We acknowledge that any static pre-trained knowledge base risks becoming outdated. However, this work focuses on **closed-book factuality alignment**, whose core objective is to address the issues of the model "knowing but not stating" or "fabricating beyond its knowledge" – that is, to **faithfully express its existing parametric knowledge**. As established in related research, the primary role of RL here is to **elicit and align** the knowledge acquired during pre-training, not to inject new knowledge. The latter is the objective of continued pre-training or RAG. We appreciate the reviewer's concern. We wish to clarify that, within the current research paradigm, alignment based on a model's internal knowledge and introducing new knowledge via RAG are widely regarded as two complementary and important research directions. Our work focuses on the former, whose core goal – ensuring truthful expression of the model's existing knowledge – holds independent and significant research value.

---

> > > ### Author Response · Authors · 2025-11-27
> > > **Authors' Rebuttal 3**
> > >
> > > 2.  **Timeliness of the Knowledge Source**: We understand the concern regarding knowledge timeliness. It is important to note that the external knowledge source used to construct reward signals for our baseline methods (e.g., GRPO+FactScore), as well as the source used for evaluating the KLCF framework, was the relatively recent **Wikipedia dump from July 2025**. The experimental results show that under this setup, KLCF performs comparably to or even better than these baselines. This indicates that for the **closed-book QA** task, providing reward signals based on internal knowledge consistency is sufficiently effective. The goal is faithful expression of the knowledge the model already possesses, rather than relying on supervision from the latest external knowledge not yet mastered by the model. Furthermore, our evaluation on key test sets like LongFact and Factory-Hard was also conducted using this July 2025 knowledge base, ensuring the accuracy of our evaluation conclusions.
> > >
> > > 3.  **Reward Model Reliability and Dual Safeguard**:
> > >     *   **Performance Verification**: As shown in Appendix Table 13, our trained truthfulness reward models achieved accuracies of 81.87%, 83.47%, and 83.34% on their respective 3000-sample open test sets for the 7B, 14B, and 32B models, showing significant improvement over the base models. We believe the primary constraint preventing further accuracy gains is precisely the inherent knowledge boundary of the base models themselves.
> > >     *   **Mechanism Safeguard**: More importantly, we do not rely solely on this single model. As mentioned in the main text, the **Checklist Reward** is based on offline-verified high-confidence facts, providing a stable "factual anchor" for training. This **dual-reward design acts as a double safeguard**, effectively preventing potential errors from a single reward model from dominating the entire training process.
> > >
> > > 4.  **Error Case Analysis**: We analyzed error cases from the reward model. Typical examples include:
> > >     *   **False Negative**: The claim "**Hurricane Donna formed on September 1, 1960**" was incorrectly judged as false. This is actually a correct fact. The error might stem from the model encountering conflicting temporal information during pre-training or failing to accurately associate the precise date with this specific meteorological event.
> > >     *   **False Positive**: The claim "**The bird louse leech feeds on the internal organs of birds**" was incorrectly judged as true. This reflects a classic error in the model's internal knowledge – potentially confusing two entirely different organisms, "bird lice" (which feed on feathers) and "leeches" (which feed on blood), and erroneously combining their characteristics into a non-existent creature and its habits.
> > >
> > > In summary, while relying solely on internal knowledge has theoretical limits, by using a recent knowledge source, validating reward model performance, and implementing a dual-reward interlocking mechanism, we have maximized the practical effectiveness and robustness of the KLCF framework.

---

> > > > ### Author Response · Authors · 2025-11-27
> > > > **Kindly Reminder**
> > > >
> > > > Dear Reviewer st4y,
> > > >
> > > > Thank you once again for your insightful and constructive feedback on our manuscript. We greatly appreciate your comments, which have helped us significantly improve the work.
> > > >
> > > > In our response, we have made substantial efforts to address your key concerns:
> > > > *   We have strengthened the experiment on the **"hallucination tax"** by fine-tuning on a mixed objective and demonstrating preserved factuality.
> > > > *   We have further clarified the **conceptual distinctions** of our dual-reward design compared to prior art.
> > > > *   We have included the requested **efficiency breakdown** and a detailed **error analysis** of the reward model.
> > > >
> > > > We hope that our revisions and detailed responses have adequately addressed your questions. As the discussion period is concluding, we would be grateful if you could let us know if you are satisfied with our responses or if any points require further clarification.
> > > >
> > > > Thank you for your time and consideration.
> > > >
> > > > Best regards,
> > > >
> > > > The Authors

---

### Author Response · Authors · 2025-11-21
**General Response 1 to Common Reviewers' Comments**

We sincerely thank all reviewers for their insightful comments and suggestions. The expert feedback has been immensely helpful and has enabled us to make comprehensive improvements and enhancements to our paper.

After carefully reviewing all comments, we have identified several key concerns raised by multiple reviewers. To present our revisions and responses clearly, we address these points collectively in this general response. Below, we provide detailed responses to each of these points.

**Q1**: **Is our KLCF framework truly efficient?**

We sincerely thank Reviewers `st4y` and `7TbV` for their valuable comments regarding computational efficiency. You rightly pointed out that our method involves offline data preparation and reward model training costs, and that a comprehensive efficiency evaluation is crucial to validate our "more efficient" claim.

To clarify this point clearly, we conducted a detailed cost analysis. Using the Qwen2.5-14B model as an example, we break down the end-to-end process time as follows:
| Phase | Key Steps | Hardware Config | Total Time | Key Notes |
| :--- | :--- | :--- | :--- | :--- |
| **A. Offline Prep (One-time)** | 1. Base Model Response Sampling 2. Claim Extraction 3. Claim Verification 4. Factual Checklist Construction 5. Truthfulness Reward Model Training 6. Checklist Verifier Training | 4 × H100 (80GB) | **~20 hours** | All outputs (checklists, reward models) are **permanently reusable** for any subsequent RL training runs, different model scales, or even different tasks. |
| **B. KLCF Online RL (Per Step)** | Policy Rollout (Response Generation) | 16 × H100 (80GB) | **~35 seconds** | This part is **identical** to RL baselines using rewards like FActScore and represents the baseline overhead of RL training. |
| | **KLCF Reward Computation** | | **~127 seconds** | Uses our lightweight, local reward models; **requires no external API calls**. |
| | **Others** | | **~50 seconds** | For example, advantage estimation and parameter updates. |
| | **KLCF Total per Step** | | **~212 seconds** | |
| **C. FActScore-based Online RL (Per Step)** | Policy Rollout (Response Generation) | 16 × H100 (80GB) | **~35 seconds** | Same as our method. |
| | **FActScore Reward Computation** | (Requires External API) | **~1056 seconds** | Relies on external search and verification, subject to network latency and API rate limits. |
| | **Others** | | **~50 seconds** |  |
| | **FActScore Total per Step** | | **~1141 seconds** | |

Based on the data above, we can draw the following key conclusions to support our efficiency claim:

1.  **Reward Computation is the Decisive Bottleneck for Training Efficiency**: As shown in the table, during online training, the reward computation time (KLCF: 127s, FActScore: 1056s) far exceeds the policy rollout time (35s). Our reward computation time is only **12%** of FActScore's, making a single training iteration **~5.38x faster** (KLCF: 212s vs. FActScore: 1141s).

2.  **One-time Offline Investment Yields Sustained and Scalable Efficiency Gains**:
    *   Although we have a one-time offline preparation cost of ~20 hours, this investment pays off quickly in subsequent training runs.
    *   **Quantitative Comparison**: Assuming 100 RL training steps, the total online time for KLCF is `100 * 212s ≈ 5.89 hours`, whereas the FActScore baseline requires `100 * 1141s ≈ 31.69 hours`. **In just a single training run, we save approximately 25.8 hours**. For large-scale training requiring thousands of steps or multiple hyperparameter tuning sessions, the accumulated time savings would be substantial, far exceeding the initial one-time offline investment.

3.  **Addresses Critical Bottlenecks in Practical Deployment**: FActScore-based baselines face severe challenges in practical large-scale RL training. Frequently making large-scale requests to external search APIs is not only slow but also subject to rate limits and network instability, becoming a critical bottleneck. Furthermore, in **mixed-task training** (e.g., combining factuality tasks with mathematical reasoning), all tasks must wait for the slowest reward signal. If one reward relies on external search, it slows down the entire mixed-task training efficiency. Our method, with its locally computed and efficient rewards, completely avoids this issue, providing a solid foundation for stable and efficient large-scale RL training.

In summary, we acknowledge the initial offline preparation cost. However, our detailed data comparison demonstrates that **our approach transforms the primary cost into a one-time, reusable investment that tackles the main efficiency bottleneck in online RL training**. Consequently, it achieves significantly higher overall efficiency, especially in large-scale training scenarios.

We thank the reviewers again for prompting us to clarify this point more explicitly. We have incorporated this analysis and the corresponding results into the revised manuscript.

---

> ### Author Response · Authors · 2025-11-21
> **General Response 2 to Common Reviewers' Comments**
>
> **Q2**: **What is the innovation of this paper?**
>
> We sincerely thank reviewers `st4y` and `XrM1` for their thoughtful questions regarding the innovation presented in this paper. We thank the reviewer for this insightful comment about the incremental nature of our contribution. We agree that using a model’s internal knowledge for factuality verification has been explored in prior work. We view KLCF as a systematic extension of this line of research, with its core contribution lying in addressing a central challenge of the post-pretraining era: how to efficiently and reliably convert a pretrained model’s vast parametric knowledge into the expressed knowledge of an aligned model.
>
> Most existing approaches tackle only one side of this challenge. For example, “self-assessment” methods mainly suppress the generation of unknown content to avoid out-of-bound hallucinations, while alignment methods based on metrics such as FActScore rely on external rewards to constrain outputs to known facts. However, these methods do not formulate consistency between parametric and expressed knowledge as a unified, bidirectional alignment objective that is jointly optimized—one that simultaneously encourages the model to fully and accurately express the knowledge it already possesses, and prevents it from fabricating content beyond its knowledge boundaries.
>
> KLCF explicitly targets this tension by proposing a unified dual-fact alignment framework. Our novelty is not a simple combination of two existing ideas. First, we design a systematic offline data pipeline to extract and verify parametric knowledge from the base pretrained model, constructing both a factual checklist and a truthfulness reward model. We then formalize the two dimensions of knowledge consistency—recall and precision—into complementary reward signals that are jointly optimized within a synergistic online RL framework.
>
> We believe this framework offers a new pathway for tackling knowledge-level consistency alignment in the post-pretraining era. Empirically, KLCF attains a superior performance across multiple factuality benchmarks, while also demonstrating efficiency and scalability in large-scale online RL settings. These results provide an initial validation of pretraining–post-training knowledge consistency as a viable and powerful alignment paradigm. In the revised manuscript, we further strengthened the discussion to more clearly articulate KLCF’s distinctions from and connections to prior work. We again thank the reviewer for this valuable feedback, which helps us improve the clarity and positioning of our contribution.

---

### Meta-Review · Area_Chair_huXc · 2026-01-05

**Summary:**

Based on the reviews and subsequent discussions, I recommend rejection of this submission. While the paper tackles an important problem and shows experimental improvements, several fundamental concerns remain unresolved despite the authors' extensive rebuttal efforts.

The substantive concern is on the contribution's novelty and rigor. Reviewer st4y is concerned that the mismatch between how "hallucination tax" is defined versus tested, and about the incremental nature of combining existing ideas (internal knowledge verification + dual rewards). Reviewer XrM1, despite not opposing acceptance, maintained their score and raised valid questions about whether the dual-reward complexity is actually necessary. The ablation in Table 2 shows removing checklist precision barely hurts performance, undermining claims about the framework's essential components. Reviewer 7TbV's final comments, while acknowledging partial improvements, introduced several new technical concerns about fair efficiency comparisons, reward variance analysis, and whether the offline preparation costs are being properly accounted for. The fact that critical experiments (like the helpfulness+factuality mixed training) only appeared during rebuttal rather than the original submission also raises questions about how thoroughly the initial work was evaluated.

The efficiency claims remain particularly problematic. While the authors added detailed breakdowns, the comparison relies on a dedicated 4×H100 GPU setup for rewards that may not reflect realistic deployment, and as 7TbV notes, the FActScore baseline could be substantially accelerated with better implementation choices. The "online external-knowledge-free" framing, though technically accurate after revision, still obscures that the offline Wikipedia-based preparation is doing much of the heavy lifting. More fundamentally, the paper hasn't convincingly demonstrated that its particular combination of checklist-based and confidence-based rewards offers advantages that justify the added system complexity over simpler shaped reward approaches.

**Reviewer Concerns:**

Looking at which concerns were genuinely resolved, Reviewer uxHw's issues were fully addressed. The authors added the requested experiments on instruction-tuned models, corrected figure labels, and provided detailed specifications for the claim extraction model with case studies. This reviewer explicitly expressed satisfaction and maintained their positive score of 6, so there are no outstanding issues here.

For Reviewer st4y, the situation is more ambiguous. The authors provided substantive responses to all points, including a new mixed-objective experiment for the hallucination tax concern, the requested Self-Eval-p(True) baseline comparison, and detailed efficiency breakdowns with error analysis. However, this reviewer never responded to the rebuttal despite being prompted by the authors. This silence is concerning because it suggests the explanations may not have been convincing enough to warrant engagement. The core concern about incremental contribution remains particularly unresolved. While the authors reframed their work as addressing knowledge-level consistency in the post-pretraining era, they didn't fundamentally address whether this constitutes a novel framework versus a competent engineering combination of existing ideas like self-assessment and checklist-based verification.

Reviewer XrM1 presents a mixed picture. The authors improved writing structure and clarified experimental details, but the fundamental question about dual-reward necessity persists. The reviewer's follow-up reveals continued skepticism, asking five detailed questions about checklist completeness, whether higher recall is always better, potential reward conflicts, and requesting concrete examples. Most tellingly, the ablation study in Table 2 shows that removing the checklist precision reward has minimal performance impact, which directly undermines the authors' claims about essential complementarity between the two factual rewards. The reviewer explicitly stated they're keeping their score of 4 while not opposing acceptance, which reads as diplomatic reluctance rather than genuine endorsement.

Reviewer 7TbV initially seemed more positive, explicitly stating they would adjust their score after concerns were partially addressed. However, their follow-up comments introduced multiple new technical concerns that suggest the rebuttal actually raised more questions than it answered. On efficiency, they pointed out that the comparison may be unfair because the 4×H100 dedicated GPU setup doesn't reflect how modern frameworks like Verl or OpenRLHF actually distribute reward computation, that the FActScore baseline could be substantially faster with local deployment rather than API calls, and that correlation data between response length and computation time is missing. On reward design, they questioned whether the fact precision and truthfulness rewards are truly functionally distinct, noting both fundamentally drive the same strategy of refusing to answer uncertain knowledge. They also requested several missing ablations including training with only recall and truthfulness rewards but zero precision weight, analysis of reward variance within GRPO groups rather than just absolute values, and experiments removing the general reward to isolate its effect on response length. The phrase "partially addressed" combined with these extensive new critiques suggests the score adjustment may not be favorable, and at minimum indicates substantial work remains to satisfy this reviewer's concerns about the method's rigor and the fairness of its experimental evaluation.

**Reviewer Scores:**

Reviewer uxHw would almost certainly maintain their score of 6. They explicitly stated in their follow-up that they were pleased to see the method works well on instruction-tuned models and that their original assessment remains appropriate. While they had a minor quibble about the claim that Qwen2.5-Base is the "purest" setting, this didn't seem to diminish their overall positive evaluation. This reviewer appears genuinely satisfied with the responses.

Reviewer st4y presents the most uncertain case precisely because they never responded despite being directly prompted by the authors. This silence after a substantive rebuttal is rarely a positive sign. If they had been satisfied, a brief acknowledgment would have been natural, especially given the authors' explicit request for feedback as the discussion period closed. The most likely scenario is they would maintain their score of 4, though there's a real possibility they might lower it to 3 if they felt the responses revealed deeper issues rather than resolving their concerns. Their fundamental worry about incremental contribution wasn't really addressed in substance, just reframed with different language about the post-pretraining era. The lack of engagement suggests either dissatisfaction with the responses or a judgment that further discussion wouldn't be productive.

Reviewer XrM1 explicitly stated their intention to keep their score of 4, saying "I lean towards keeping my score" while diplomatically adding they "wouldn't mind if the paper gets accepted." This is not an enthusiastic endorsement. The five detailed follow-up questions they posed, particularly about reward conflicts and checklist completeness, indicate genuine remaining concerns rather than satisfied curiosity. The fact that they characterized the paper as "interesting" but still maintained their below-threshold score after the rebuttal suggests they found the responses informative but not sufficiently convincing to change their fundamental assessment. They would almost certainly stay at 4.

Reviewer 7TbV explicitly said they would "adjust my score accordingly" after concerns were "partially addressed," but the direction of this adjustment is genuinely unclear and could go either way. On one hand, they acknowledged improvements on the terminology and related work issues, which might suggest a bump to 5 or even 6 if they're crediting the authors' effort and responsiveness. On the other hand, their extensive follow-up comments introduced multiple new technical concerns about efficiency comparison fairness, missing ablations, reward variance analysis, and GPU framework details. The tone of these follow-ups seems more critical than satisfied, with phrases like "I believe a more comprehensive discussion is needed" and questioning whether comparisons are "entirely fair." The fact that the rebuttal apparently surfaced new problems rather than simply resolving old ones could just as easily lead to maintaining the 4 or even dropping to 3 if they concluded the work is less rigorous than initially thought. Given the ambiguous phrasing and the substantive nature of the new critiques, a reasonable guess would be they might raise to 5 as a gesture toward the authors' efforts, but this is far from certain.

---

### Decision · Program_Chairs · 2026-01-26

Reject